# Implicit Inversion turns CLIP into a Decoder

**Antonio D'Orazio**[1]   **Maria Rosaria Briglia**[1]   **Donato Crisostomi**[2,1]   **Dario Loi**[1]
**Emanuele Rodolà**[2,3]   **Iacopo Masi**[1]

Sapienza University of Rome, Italy       [1]OmnAI Lab 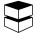   [2]GLADIA   [3]Paradigma

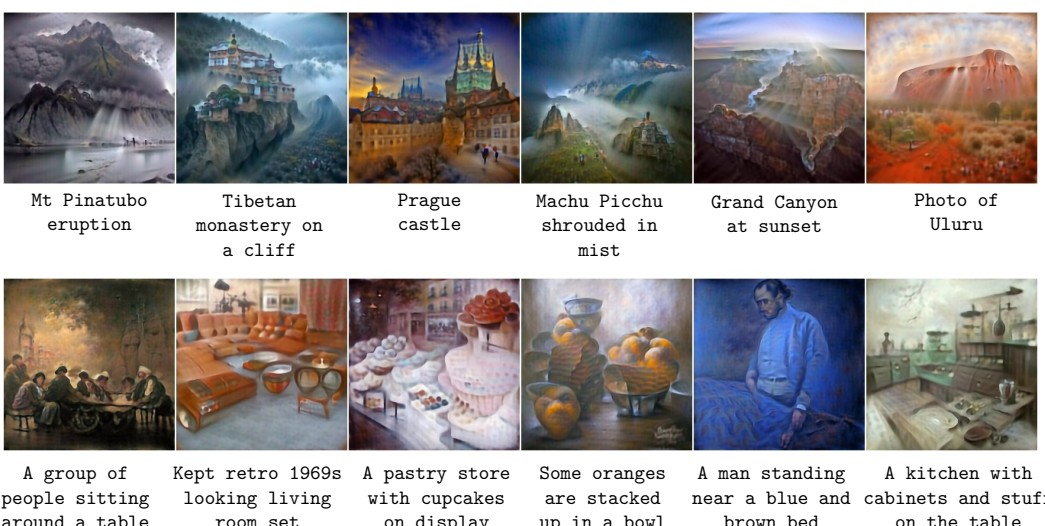

Figure 1: **Text-to-image synthesis by CLIP inversion.** CLIP$^{-1}$ inverts CLIP's image encoder using implicit neural representations, enabling text-to-image synthesis *without any fine-tuning of CLIP or dedicated generative decoder.* All samples are generated with CLIP ViT-B/32 Radford et al. (2021), with *top* rows showing generic scene prompts and *bottom* rows illustrating complex captions from MS-COCO Lin et al. (2014).

## Abstract

CLIP is a discriminative model trained to align images and text in a shared embedding space. Due to its multimodal structure, it serves as the backbone of many generative pipelines, where a decoder is trained to map from the shared space back to images. We show that image synthesis is nevertheless possible using CLIP alone—without a pre-trained generative decoder or CLIP tuning. Our approach optimizes a frequency-aware implicit neural representation that encourages coarse-to-fine generation by stratifying frequencies across network layers. To stabilize this inverse mapping, we introduce adversarially robust initialization, a lightweight Orthogonal Procrustes projection to align local text and image embeddings, and a blending loss that anchors outputs to natural image statistics. With CLIP frozen, this framework unlocks capabilities such as text-to-image generation, style transfer, and image reconstruction. Our findings suggest that discriminative models may hold untapped generative potential, hidden in plain sight. Code: https://github.com/OmnAI-Lab/implicit-inversion.

## 1 Introduction

Text-to-image generation has progressed from a challenge to a widely accessible technology, with recent models achieving photorealistic results and remarkable creative abilities (Betker et al., 2023; Saharia et al., 2022; Nichol et al., 2022; Chang et al., 2023). Among these, latent diffusion models (Rombach et al., 2022) currently define the state-of-the-art by relying on an encoder–decoder architecture, where the encoder maps the input text into a latent representation, and the decoder reconstructs an image from it. Foundational vision-language models like CLIP (Radford et al., 2021)

are often used as text encoders (Betker et al., 2023; Wang et al., 2022; Tao et al., 2023), whereas the decoder is typically a diffusion model – by far one of the most computationally demanding stages.

In this paper, *we show that CLIP alone can perform text-to-image generation, without a pretrained generative decoder.* We achieve this by inverting the CLIP vision encoder: rather than mapping an image to its latent embedding, we start from a CLIP embedding and reconstruct the corresponding image. Although prior work has attempted this through direct pixel space optimization (Kazemi et al., 2024; Ganz & Elad, 2023; 2024), such approaches produce low-quality output with visible artifacts (Kazemi et al., 2024) or require additional training (Ganz & Elad, 2023; 2024). In contrast, we introduce CLIP$^{-1}$, a *pretrained-decoder-free and CLIP-tuning-free* solution based on Implicit Neural Representations (INRs) (Liu et al., 2024). Fig. 1 offers our results on generic prompts and longer, more complex prompts from MS-COCO (Lin et al., 2014).

We first retrieve a low-frequency seed INR whose caption is most similar to the prompt. Because this INR was pre-trained on a blurred version of its image with Adversarial Weight Perturbations (AWP) (Wu et al., 2020), its low-frequency weights are stable and provide a robust anchor. We then refine the INR layer by layer: a peaked learning-rate schedule moves from low- to high-frequency layers, progressively adding details in a coarse-to-fine manner. During refinement, we encourage on-manifold solutions using cosine losses against a style prompt and a few retrieved natural images, eventually resulting in more aesthetic generations. Without pixel-space optimization, our approach avoids structural artifacts and results in a coarse-to-fine generation akin to diffusion models.

Despite the stability of the INR backbone, inversion still suffers from CLIP's modality gap (Liang et al., 2022b; Mistretta et al., 2025; Zhang et al., 2024): text and image embeddings fall on slightly offset sub-manifolds, making the raw prompt embedding an unreliable target. We bridge this gap by seeking an optimal orthogonal transformation via Procrustes analysis (Wang & Mahadevan, 2008; Maiorca et al., 2023), using the $k$ nearest caption–image pairs to align the prompt embedding with the image manifold and produce a well-conditioned target for optimization.

We benchmark our pipeline on 10k MS-COCO (Lin et al., 2014) captions, reporting Fréchet Inception Distance (FID) (Heusel et al., 2017), Inception Score (IS) (Salimans et al., 2016), and CLIPSIM metric (Hessel et al., 2021). Compared with DAS (Fort & Whitaker, 2025), a concurrent inversion-based approach that similarly requires no pretrained decoder or CLIP fine-tuning, our method achieves half the FID while almost doubling IS, producing crisper, more faithful images. The same frozen model transfers zero-shot to image reconstruction, controlled image edits, and neural style transfer, confirming the versatility of the INR backbone. Ablations show that *(i)* adversarially robust INRs boost quality, *(ii)* Procrustes alignment yields sharper, more semantically aligned images, and *(iii)* frequency-steered optimization suppresses high-frequency artifacts.

In summary, our contributions are: (1) **Text-to-image synthesis by inversion, without pretrained decoders or CLIP tuning.** We repurpose a *frozen* CLIP as a generator by optimizing a **frequency-aware INR**, retrieved via a blur-initialized anchor and refined through a coarse-to-fine schedule driven only by CLIP losses, without external decoders or re-training. (2) **Modality-gap reduction** We reduce the mismatch between CLIP's text and image sub-manifolds through an orthogonal Procrustes transformation estimated from the $k$ nearest caption–image pairs. (3) **Extensive empirical validation.** We evaluate reconstruction, controlled edits, style transfer, and interpretability tasks. We release all code and models to support future research.

We emphasize that our goal is not to compete with diffusion or autoregressive decoders in fidelity, but to show that *sufficiently realistic* images can be generated using only CLIP's latent space, without any external decoder. This also gives the method analytical value: it reveals how much visual structure is already encoded in a frozen CLIP and how the model reacts to challenging or atypical textual inputs, offering a practical tool for interpretability, stress-testing, and debugging multimodal systems before CLIP's errors and/or biases propagate into larger pipelines.

## 2 RELATED WORK

**Image generative models.** Modern image generators can be categorized into four broad families: GANs, diffusion models, normalizing flows, and autoregressive (AR) models. GANs train a generator to fool a discriminator with realistic samples, a strategy refined from early DCGANs (Radford et al., 2015) to StyleGAN-T (Sauer et al., 2023) and BigGAN (Brock et al.). Diffusion models be-

gin with pure noise and iteratively denoise it back to an image; latent diffusion (Rombach et al., 2022), GLIDE (Nichol et al., 2022), DALL-E 2 (Ramesh et al., 2022) and DALL-E 3 (Betker et al., 2023) differ mainly in how they compress the signal and guide it with text. Normalizing flows (e.g., (Kingma & Dhariwal, 2018; Esser et al., 2024)) learn a chain of invertible transforms so that sampling the base Gaussian and running the reverse pass yields data in a few steps. AR models play a significant role in the natural language processing domain, with a decoder-only transformer (Vaswani et al., 2017) powering tools such as LLMs. Although AR models have been primarily applied in discrete text domains, Tian et al. (2024) demonstrates that they also yield state-of-the-art generation performance in the image domain, competing with diffusion models. Despite these distinct mechanics, every approach relies on a latent-to-image decoder: the generator in GANs, the denoising network in diffusion, or the reverse flow in normalizing-flow models, along with the internal representation in transformers in AR models. Our work bypasses the need for a pretrained latent-to-image decoder by directly inverting a frozen discriminative encoder (CLIP).

**Model inversion.** Several recent works attempt to repurpose CLIP for image generation by inverting its embeddings. The earliest approach (Kazemi et al., 2024) directly optimises randomly initialised pixels to minimise the cosine distance between the CLIP embedding of the image and that of a target text prompt. GALIP (Tao et al., 2023) introduces a CLIP-conditioned GAN framework, training both a generator and discriminator to enable fast, controllable synthesis with fewer parameters and less data than large-scale diffusion models. CLIPAG (Ganz & Elad, 2023) does not rely on a pretrained decoder but applies adversarial fine-tuning to CLIP itself to improve generation quality. EB-CLIP (Ganz & Elad, 2024) further extends this idea, where the generation is formulated as energy minimisation in CLIP's joint image-text space. Concurrently with our work, DAS (Fort & Whitaker, 2025) shows that a frozen CLIP can be inverted by optimising at multiple spatial resolutions in a coarse-to-fine manner. Like us, DAS reveals generative priors within discriminative models, but differs by operating directly in pixel space rather than through a frequency-aware implicit representation.

**Adversarial robustness.** Adversarial training has become a core strategy for improving model robustness, with (Madry et al., 2018) introducing the first widely adopted method using input perturbations. TRADES (Zhang et al., 2019) extended this by leveraging KL divergence to balance accuracy and robustness, later refined by Cui et al. (2023) to address its asymmetry. Beyond input-space attacks, AWP (Wu et al., 2020) proposed perturbing model weights during training, improving generalisation by flattening the loss landscape. While primarily used for classification, recent work (Mirza et al., 2024) suggests robust models also encode stronger generative priors.

**Image modeling with INRs.** Implicit Neural Representations model images as continuous functions that map spatial coordinates $(i, j)$ to RGB values via a neural network, typically an MLP. To capture fine detail, they rely on frequency-aware components such as positional encodings (Tancik et al., 2020) or periodic activation functions like SIREN (Sitzmann et al., 2020). However, fixed-frequency activations limit adaptability, motivating FINER (Liu et al., 2024), which introduces variable-periodic activations that dynamically adjust to local frequency content. We adopt FINER for its efficient representation, well-suited to our inversion task, and to mitigate the spectral bias that hampers high-fidelity reconstruction in standard INRs.

## 3 METHOD

Given an input text prompt, our approach aims to generate images by inverting the corresponding CLIP embeddings. The pipeline consists of three main stages: *(i)* a *data preparation* step (performed once offline), *(ii)* an *initialization* step leveraging preprocessed data to retrieve a suitable starting point, *(iii)* an *optimization* procedure refining the initial image to match the target text.

### 3.1 PRELIMINARIES

**Implicit Neural Representations.** INRs represent images as functions mapping spatial coordinates $(i, j)$ to RGB values $f_\phi(i, j) = (r, g, b)$, where $\phi$ are parameters of a neural network. The architecture is typically a Multilayer Perceptron (MLP), using positional encoding or frequency-aware activations for compact, efficient representation, crucial for image synthesis via iterative optimization. We adopt FINER (Liu et al., 2024), which enhances fine-detail modeling using variable periodic

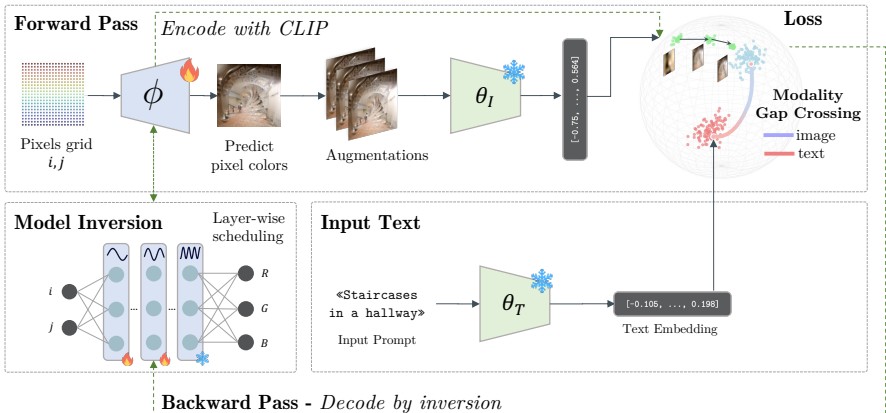

Figure 2: **CLIP$^{-1}$ text-to-image inversion pipeline**. The image is represented with an Implicit Neural Representation (INR) $f_\phi(i, j)$, optimizing weights to match an input text prompt. The inversion starts from a robust INR trained with Adversarial Weight Perturbation (AWP). The optimization updates the INR layer-wise so that the embedding of its rendering aligns with the text prompt embedding. The procedure includes augmentations, CLIP embeddings averaging, and projection onto the unit sphere. To align text and image embeddings, we apply an orthogonal Procrustes transformation to address the modality gap in CLIP.

activations. It is based on SIREN (Sitzmann et al., 2020), which uses fixed-frequency activation $\mathbf{z}_i = \sin\left(\omega(\mathbf{W}izi - 1 + \mathbf{b}_i)\right)$. FINER improves adaptability by introducing an additional coefficient:

$$\mathbf{z}_i = \sin\left(\omega\alpha_i(\mathbf{W}_i\mathbf{z}_{i-1} + \mathbf{b}_i)\right), \quad \alpha_i = |\mathbf{W}_i\mathbf{z}_{i-1} + \mathbf{b}_i| + 1, \tag{1}$$

where $\alpha_i$ dynamically sets local frequency based on input magnitude. We leverage FINER's bias initialization, which stratifies frequencies across layers, capturing low frequencies early and high frequencies deeper, leading to better convergence and reconstruction. See Fig. 6 for an illustration.

## 3.2 DATA PREPARATION

The computation of the CLIP embedding of both images and captions of the chosen dataset is performed offline once and stored in a lightweight index using FAISS (Johnson et al., 2019), which is then used to retrieve the dataset sample closest to the input prompt. We used images from LAION Aesthetics, a subset of LAION-5B by Schuhmann et al. (2022). This data serves two purposes: *(i)* training INRs for initializing text-to-image optimization and *(ii)* computing natural image CLIP embeddings to act as anchor points in the latent space.

Each image is first blurred using a Gaussian filter to suppress high frequencies, providing a smoother starting point for inversion. An INR is then trained to reconstruct the blurred image, and its weights are stored. For the $i$-th image, the predicted pixel values from the INR are $f_{\phi^i}$, with weights $\phi^i$. The INR images are then encoded via CLIP to obtain visual embeddings $\boldsymbol{\theta}_I\left(f_{\phi^i}\right)$. The CLIP text embeddings $\boldsymbol{\theta}_T\left(\mathbf{y}^i\right)$ of the corresponding captions are also stored. This results in a dataset $\mathcal{D} = \{\boldsymbol{\theta}_I\left(f_{\phi^i}\right), f_{\phi^i}, \boldsymbol{\theta}_T\left(\mathbf{y}^i\right)\}$, containing CLIP image embeddings, INRs, and CLIP text embeddings for each training sample.

**Robust INR initialization.** The INR weights $\phi^i$ can be viewed as representations of the $i$-th image – each uniquely capturing its content, like the RGB representation in pixels. However, small perturbations in these weights can significantly alter the reconstructed image, making them sensitive and potentially unstable for downstream tasks. To address this, we propose a training method that improves INR weights robustness by incorporating adversarial perturbations during training. Given an INR with weights $\phi$, we define an adversarial weight perturbation (AWP) $\Delta\phi \in \Omega$, where $\Omega$ bounds the perturbation range. The training objective is to make the model resilient to such perturbations by solving the following min-max optimization:

$$\min_\phi \max_{\Delta\phi \in \Omega} \mathcal{L}\left(f_{\phi+\Delta\phi}, \mathrm{blur}(\mathbf{x})\right), \tag{2}$$

where $f_{\phi+\Delta\phi}$ is the perturbed INR output and $\mathcal{L}$ is the reconstruction loss function w.r.t. the blurred target image $\mathbf{x}$. Unlike standard adversarial training in the input space (Wu et al., 2020), our approach perturbs only the model weights. The perturbation set is constrained by a relative norm bound $\Omega = \{\Delta : \|\Delta\| \leq \gamma\|\phi\|\}$, with $\gamma$ controlling the allowed perturbation magnitude. This robust training is applied only once, during the construction of the initial INR at the inversion step $n = 0$. It ensures that early optimization steps do not cause the weights to drift too far from the frequency content of the initialization, improving stability during inversion. The supplementary material details the training procedure using the AWP algorithm (§A.3). Since training a single INR is fast, it can be fit on the fly, eliminating the need to store weights for the entire dataset. The resulting weights can be cached and reused for reproducibility and faster consecutive generations with the same initialization, or discarded to encourage more diverse samples.

## 3.3 INITIALIZATION AND MODALITY GAP HANDLING

**Inversion initialization.** Text-to-image generation begins with a text prompt $\mathbf{y}$, which is encoded using the CLIP text encoder to obtain $\mathbf{e}_t = \boldsymbol{\theta}_T(\mathbf{y}) \in \mathbb{R}^d$. To initialize the inversion, we search our dataset $\mathcal{D}$ for the INR whose associated caption has the highest cosine similarity to $\mathbf{e}_t$. This serves as the starting point for the optimization.

**Bridging the modality gap.** CLIP aligns text and image embeddings globally, projecting them onto the unit sphere. However, local differences between modalities persist: text embeddings tend to encode abstract semantics, while image embeddings reflect concrete visual features. Directly optimizing an image to match a text embedding often causes artifacts or *textual hallucinations* – the model overfits to abstract concepts and produces unrealistic visuals (Liang et al., 2022a).

To address this, we learn a local transformation to align text and image embeddings more precisely. We retrieve the $k$ nearest neighbors of the input text embedding $\mathbf{e}_t$ from our dataset $\mathcal{D}$, forming two matrices: $\mathbf{E}_T \in \mathbb{R}^{d \times k}$, encoding the $k$ closest text embeddings, and $\mathbf{E}_I \in \mathbb{R}^{d \times k}$ encoding their corresponding image embeddings. Solving the orthogonal Procrustes problem we find an orthogonal matrix $\mathbf{R}$ that best aligns these two sets:

$$\min_{\mathbf{R}} ||\mathbf{R}\mathbf{E}_T - \mathbf{E}_I||_F \qquad \text{s.t. } \mathbf{R}^\top\mathbf{R} = \mathbf{I}, \tag{3}$$

where $\|\cdot\|_F$ is the Frobenius norm. This is performed for each input prompt. The resulting orthogonal matrix $\mathbf{R}$ aligns the local structure of text embeddings with that of image embeddings. We then transform the input text embedding into the image modality as $\mathbf{e}_{t2i} = \mathbf{R}\mathbf{e}_t$, which becomes the target embedding for the CLIP inversion process.

## 3.4 INVERTING CLIP WITH IMPLICIT NEURAL REPRESENTATIONS

**Text-To-Image via CLIP Inversion.** Our pipeline, shown in Fig. 2, inverts a CLIP text embedding to synthesize an image using an INR. Given a text prompt $\mathbf{y}$, we obtain the projected image-space embedding $\mathbf{e}_{t2i} = \mathbf{R}\boldsymbol{\theta}_T(\mathbf{y})$. The INR $f_\phi$ is optimized so that its output image matches $\mathbf{e}_{t2i}$ when passed through the frozen CLIP image encoder $\boldsymbol{\theta}_I$. This is formalized as:

$$\phi = \arg\min_\phi \mathcal{L}(\boldsymbol{\theta}_I(f_\phi), \mathbf{e}_{t2i}) \quad \text{where} \quad \mathbf{e}_{t2i} = \mathbf{R}\boldsymbol{\theta}_T(\mathbf{y}). \tag{4}$$

Here, $\mathcal{L}$ is the cosine distance, and gradients flow from CLIP back to the INR parameters $\phi$.

**Layer-wise frequency optimization.** Instead of optimizing pixel values , we update the INR weights, leveraging FINER's property that network layers correspond to different frequency bands. The INR is structured as an $L$-layer MLP, with each layer representing a specific frequency range. To guide the optimization process, we apply Gaussian learning rate scheduling: at each iteration, we focus the optimization on a specific layer by assigning it a peak learning rate, while attenuating the rates of neighboring layers according to a Gaussian curve (see Fig. 7 in the Appendix). This helps reconstruct coarse features before fine details, improving stability and fidelity.

**Augmentations for stable optimization.** Following CLIPDraw (Frans et al., 2022) and CLIPAG (Ganz & Elad, 2023), we apply color, scale, and shear augmentations during optimization. Augmentations are CLIP-encoded, averaged, and projected onto the unit sphere: $\mathbf{e}_i^\star = \frac{1}{n}\sum_{k=1}^n \boldsymbol{\theta}_I(\text{augment}(f_{\phi^k}))$, enforcing robustness to distortions.

Table 1: **MS-COCO text-to-image generation results**. FID (lower is better), CLIPSIM and IS (both higher are better), along with model sizes.

| | Betker et al. (2023) DALL-E | Nichol et al. (2022) GLIDE | Rombach et al. (2022) LDM-KL-8 | Ganz & Elad (2023) CLIPAG ViT | Ganz & Elad (2024) EB-CLIP ViT | Ganz & Elad (2024) EB-CLIP XXL | Kazemi et al. (2024) CLIP-Inv ViT | Fort & Whitaker (2025) DAS ViT | Fort & Whitaker (2025) DAS Ensemble | **Ours** $\mathbf{CLIP^{-1}}$ |
|---|---|---|---|---|---|---|---|---|---|---|
| Synthesis by inversion | ✗ | ✗ | ✗ | ✓ | ✓ | ✓ | ✓ | ✓ | ✓ | ✓ |
| Tuning-free | ✗ | ✗ | ✗ | ✗ | ✗ | ✗ | ✓ | ✓ | ✓ | ✓ |
| # Train params (M) | 12000 | 6000 | 1450 | 88 | 88 | 1200 | 0 | 0 | 0 | **0** |
| # Tot params (M) | 12000 | 6000 | 1450 | 150 | 150 | 846 | 150 | 150 | 3x150 | **150** |
| FID(↓) | 27.5 | 12.2 | 23.3 | 42.3 | 68.3 | 23.4 | 140.1 | 161.8 | 121.6 | **72.5** |
| CLIPSIM(↑) | – | – | – | 34.7 | 34.5 | 33.5 | **61.4** | 22.7 | 36.9 | 38.6 |
| IS(↑) | 17.9 | – | 20.03 | 18.7 | – | – | 4.8 | 5.7 | 8.36 | **9.5** |

**Blending natural image priors.** To further guide generation toward realistic outputs, we incorporate information from natural images. For a given prompt $\mathbf{y}$, we retrieve the $k$ most similar image embeddings from a reference dataset using cosine similarity. These are linearly combined into a blended target embedding $\mathbf{e}_{img}^{\star}$, with weights given by the softmax of the similarity scores. A blending loss $\mathcal{L}_{\text{blend}}$ then encourages the output embedding $\mathbf{e}_i^{\star}$ to remain close to the manifold.

**Final optimization formulation.** The complete $\text{CLIP}^{-1}$ optimization pipeline updates the INR parameters $\phi$ to generate realistic images, leveraging both augmented embeddings and natural image priors, with no CLIP retraining or modification. The full procedure is described in Eq. (5)

$$\begin{cases} \text{(a)} & \mathbf{e}_i^{\star} = \frac{1}{n} \sum_{k=1}^{n} \boldsymbol{\theta}_I \big( \text{augment}(f_{\phi^k}) \big) \\ \text{(b)} & \phi_0 = \min_{\phi} \max_{\Delta\phi \in \Omega} \mathcal{L} \big( f_{\phi+\Delta\phi}, \text{blur}(\mathbf{x}) \big) \\ \text{(c)} & \phi_n = \phi_{n-1} - \nabla_{\phi} \Big[ \mathcal{L}(\mathbf{e}_i^{\star}, \mathbf{e}_{t2i}) + \beta \mathcal{L}_{blend} \big( \mathbf{e}_i^{\star}, \mathbf{e}_{img}^{\star} \big) \Big] \end{cases} \quad (5)$$

Step (a) computes the CLIP embedding from the augmented INR outputs; step (b) initializes the INR with adversarial weight perturbations to enhance robustness, and (c) updates the INR weights via backpropagation using both alignment and blending losses.

## 4 EXPERIMENTS

We evaluate our inversion pipeline across a range of tasks. We begin with text-to-image generation on MS-COCO (Lin et al., 2014), comparing against both standard generative models and prior inversion-based approaches, along with experiments to prove robustness to out-of-distribution samples (§4.1). Next, we demonstrate the generality of our method through zero-shot transfer to downstream tasks, including reconstruction, controlled modification, and style transfer (§4.2). Lastly, we quantify the effect of each component through an ablation study (§4.3). Additional experiments in the Appendix (§A.1) further validate the generative behavior of $\text{CLIP}^{-1}$, showing consistent alignment while capturing natural pixel-level variability across runs.

### 4.1 TEXT-TO-IMAGE SYNTHESIS BY INVERSION

**Setting.** The goal is to generate visually realistic images that are semantically aligned with a natural language description. To assess the visual fidelity of our method, we compute the Fréchet Inception Distance (FID) (Heusel et al., 2017) and the Inception Score (IS) (Salimans et al., 2016) over a subset of $10,000$ captions from MS-COCO (Lin et al., 2014). Since these metrics do not capture semantic alignment with the prompt, we also report CLIPSIM (Hessel et al., 2021), which measures the cosine similarity between the CLIP embeddings of generated images and their corresponding captions. We compare our method against prior and concurrent inversion-based approaches, including those that require CLIP fine-tuning (Ganz & Elad, 2023; Ganz et al., 2023) and those that do not (Fort & Whitaker, 2025; Kazemi et al., 2024). As our method uses a frozen CLIP and no pretrained decoder, we regard tuning-free baselines as the most relevant points of comparison. For completeness, we also report results from state-of-the-art generative models that rely on both training and a dedicated decoder (Betker et al., 2023; Nichol et al., 2022; Rombach et al., 2022). The implementation details can be found in the supplementary material.

**Results.** Table 1 reports our results along with model sizes and architectural requirements. Among inversion-based methods without pretrained decoders or CLIP tuning, $\text{CLIP}^{-1}$ achieves the lowest

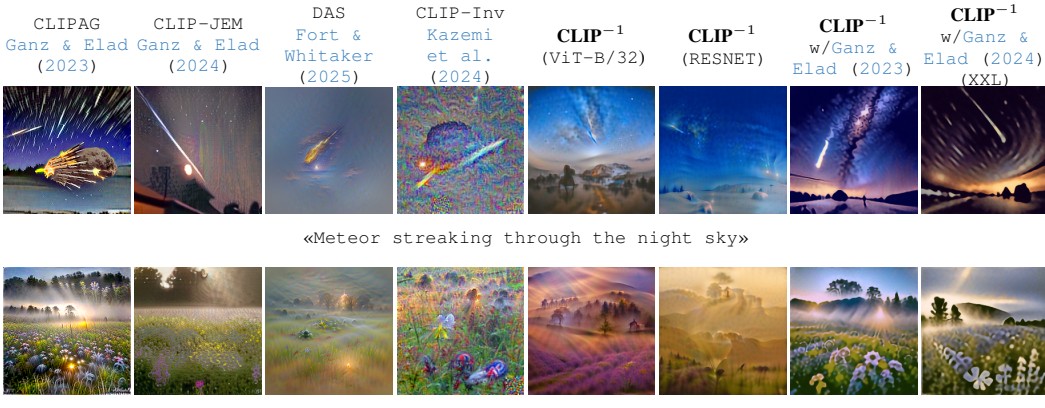

Figure 3: **Qualitative comparison** of prior pixel-based methods against different **CLIP⁻¹** configurations.

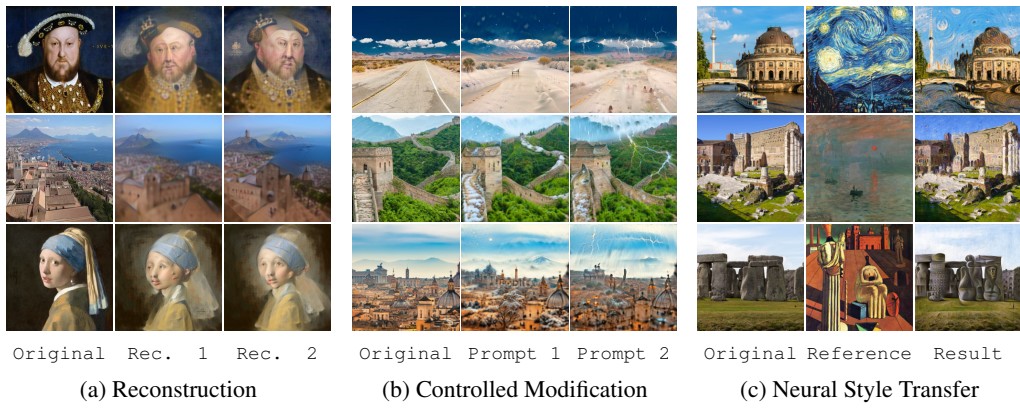

| (a) Reconstruction | (b) Controlled Modification | (c) Neural Style Transfer |

Figure 4: **Downstream Tasks.** (a) Recreates the input image from its CLIP encoding. (b) Alters the input image based on a specified prompt. Prompt 1: *«Snowy peaceful landscape»* ; Prompt 2: *«Torrential rainfall, lightning bolts»*. (c) Applies the visual style of a reference image to the input.

FID (72.5 vs. 161.8 for DAS-ViT (Fort & Whitaker, 2025)) and highest IS (9.5 vs. 5.7), marking a substantial improvement in visual quality. Although diffusion-based models still attain lower FID, they require orders of magnitude more parameters and full training pipelines, whereas our method uses a frozen backbone and a lightweight INR. Finally, our method achieves a CLIPSIM of 38.6, outperforming both training-free and fine-tuned baselines—except for CLIPInvert (Kazemi et al., 2024), whose higher CLIPSIM can be attributed to overfitting to the target embedding, as evidenced by its elevated FID and low IS. Taken together, these results indicate that Procrustes alignment and frequency-aware INR optimization effectively improve text-image consistency without modifying CLIP's weights. Figure 3 further provides qualitative examples: compared to other training-free baselines, our generations exhibit fewer structural artifacts and sharper details, and visually approach the quality of tuned approaches (Ganz & Elad, 2023; 2024). We also apply our pipeline in a plug-and-play fashion to tuned CLIP variants, such as CLIPAG (Ganz & Elad, 2023) and CLIP-JEM (Ganz & Elad, 2024), showing broader compatibility of our method with discriminatively trained models. Additional results are available in the Appendix in Figure 8.

**Robustness to distribution shift.** We ablate initialization, Procrustes alignment, and blending loss, forcing the *Plain CLIP⁻¹* variant to start from random INR weights. Despite this, it consistently outperforms the DAS baseline on MS-COCO and Flickr30k (Table 3a). Since all init/anchor embeddings derive from LAION-Aesthetics, these benchmarks constitute a strict OOD evaluation. Both Plain and Full CLIP⁻¹ retain strong performance under this shift, with initialization serving only as an optimization accelerator. Cross-dataset Fréchet distances (Table 2) confirm COCO and Flickr lie far from LAION in CLIP space, underscoring the OOD nature of these benchmarks.

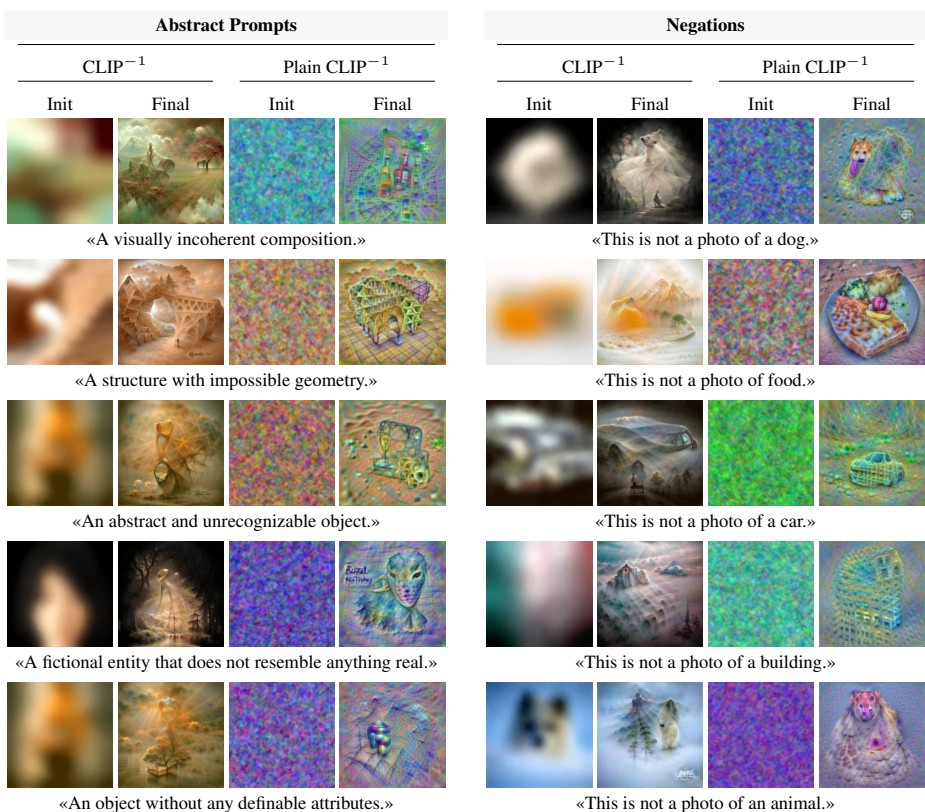

Figure 5: **Explainability of CLIP.** Exploring CLIP's behavior under out-of-distribution concepts.

## 4.2 ZERO-SHOT TASK GENERALIZATION

To assess the versatility of our approach, we explore several downstream tasks, demonstrating that the same inversion framework can successfully generate images in different settings without requiring task-specific modifications or additional optimization.

**Image reconstruction.** The goal is to reconstruct a given input image using our inversion pipeline. We treat the image as a target whose CLIP embedding is known, and optimize an INR to produce an output that matches this embedding. The INR is initialized from a blurred version of the image, and refined to align with the embedding of

Table 2: **Cross-dataset Fréchet Distances (mean ± std, 10 runs)** on CLIP embeddings of 30k images and captions. High distances confirm COCO and Flickr lie outside LAION's embedding space. Numbers in $10^{-3}$.

| Metric | Images | Caption |
|---|---|---|
| Baseline (LAION split) | $3.4 \pm 0.1$ | $6.8 \pm 0.1$ |
| Baseline (COCO split) | $4.2 \pm 0.1$ | $3.4 \pm 0.1$ |
| Baseline (Flickr split) | $4.0 \pm 0.1$ | $4.0 \pm 0.1$ |
| **COCO vs. LAION** | $365.3 \pm 0.7$ | $463.9 \pm 0.7$ |
| **Flickr vs. LAION** | $366.9 \pm 0.9$ | $503.4 \pm 1.2$ |

the full-resolution input, effectively operating as a decoder recovering semantic content from latent space. Unlike text-to-image generation, this task provides well-defined ground truth and serves as a controlled setting to evaluate inversion precision. Figure 4a shows qualitative results on both artistic and photographic inputs. Across all examples, high-level semantic content, such as facial identity or scene composition, is consistently preserved. Fine-grained spatial details, especially in structured regions like faces or buildings, are approximated with some distortion or shift, reflecting the inherent ambiguity of CLIP's embedding space.

**Controlled image modification.** The goal is to modify an input image according to a natural language prompt that specifies a targeted change in content or style. The image is first encoded via an INR fitted to its original form. A text prompt is then provided to guide the modification (e.g., *"snowy landscape"* or *"torrential rainfall"*). The INR is optimized to align the CLIP embedding of the generated image with that of the prompt, while starting from the original image representation. This setup encourages localized, semantically consistent transformations without disrupting the broader

structure or identity of the scene. Figure 4b shows three examples for the task. In each row, the left-most column is the original image; the next two columns show the edits for "snow" and "storm". The road, the Great Wall, and the city keep their geometry and colour palette, while only the requested weather effects (snow cover, rain streaks, lightning) are added. This confirms that $CLIP^{-1}$ can act as a prompt-driven image editor, producing targeted edits without explicit masks or additional training.

**Neural style transfer.** We supply two images: a *content* photo and a *style* reference. The content image is represented by an INR initialized to exactly reproduce the original photo; the style image is fed only through the frozen CLIP encoder. Optimization minimizes a weighted sum of two CLIP-based losses: *(i)* a *style loss* that pulls the INR's embedding towards that of the reference painting, and *(ii)* a *content loss* (weight 0.5) that keeps the embedding close to the original photo. Figure 4c illustrates the outcome. In each case, the brush-stroke texture and overall palette of the reference painting are transferred, while object layout and scene geometry remain intact. The method therefore separates appearance from semantics without hand-crafted losses or additional training, indicating that the inversion pipeline can exploit CLIP's latent space to disentangle style from content.

**Explainability.** Prior work shows that negative prompts and out of distribution prompts can reveal how CLIP structures semantic space and separates in distribution concepts from everything else Li et al. (2024); Nie et al. (2024). We offer a complementary view: we use our pipeline to directly visualize how CLIP interprets such texts. We evaluate two types of out of distribution targets: negative prompts such as "this is not a photo of a", and generic out of distribution prompts such as "an abstract and unrecognizable object". These reconstructions allow us to inspect how CLIP responds to these prompts and to identify which visual cues it treats as discriminative or unstable outside its training distribution. Results are shown in Fig. 5.

### 4.3 ABLATION STUDY

| Model | MS-COCO | | | Flickr30k | | |
|---|---|---|---|---|---|---|
| | FID↓ | CSIM↑ | IS↑ | FID↓ | CSIM↑ | IS↑ |
| DAS (ViT) | 161.8 | 22.7 | 5.7 | 220.5 | 38.1 | 6.5 |
| DAS (Ens.) | 121.6 | 36.9 | 8.3 | 161.1 | 39.3 | 7.1 |
| **Plain CLIP$^{-1}$** | 92.7 | **46.9** | **9.5** | 119.2 | **44.4** | 6.6 |
| **CLIP$^{-1}$** | **72.5** | 38.6 | **9.5** | **86.4** | 41.1 | **7.3** |

(a) Out of distribution evaluation

| Variant | FID↓ | CSIM↑ | IS↑ |
|---|---|---|---|
| *i.* **CLIP$^{-1}$** | **107.1** | 38.8 | 7.7 |
| *ii.* w/o Freq. Opt (F.O.) | 185.1 | 30.5 | 7.8 |
| *iii.* w/o AWP | 121.0 | 43.0 | 7.3 |
| *iv.* w/o F.O. & Proc. | 111.3 | 46.4 | **9.1** |
| *v.* w/o F.O. & $\mathcal{L}_{blend}$ | 119.7 | **49.5** | 7.9 |

(b) Quantitative ablation study

| Steps → | 40 | 400 |
|---|---|---|
| FID↓ | 107.1 | **72.5** |
| CSIM↑ | 17.8 | **38.6** |
| IS↑ | **10.6** | 9.5 |

(c) Eval. vs # steps

Table 3: (a) Out of distribution evaluation on MS-COCO/Flickr30k. Even without initialization, $CLIP^{-1}$ surpasses DAS across all metrics. (b) Quantitative ablation study on on 1,000 MS-COCO captions; legend– *i.* full model – *ii.* frequency scheduling – *iii.* AWP – *iv.* F.O. + Procrustes – *v.* F.O. + blending loss. (c) Ablations wrt to the number of steps in the inversion.

**Ablation for the proposed components.** We now perform a controlled ablation over the four key components: layerwise frequency scheduling, adversarial weight perturbation (AWP), orthogonal Procrustes alignment, and the natural-image blending loss; the corresponding ablated variants are referred to as *(i)*, *(ii)*, *(iii)*, *(iv)*. We run every variant in the same 1000 captions from MS-COCO and report FID (Heusel et al., 2017), CLIPSIM (Hessel et al., 2021), and IS (Salimans et al., 2016) in Table 3b. Better FID/IS and higher CLIPSIM show better perceptual realism and stronger text–image agreement, respectively. In parallel, we visualize representative generations so that the numerical shifts can be linked to visual outcomes. More results can be found in the supplementary material.

The ablation shows that each proposed component plays a distinct, complementary role (see Fig. 9). When the layer-wise learning-rate schedule is removed *(ii)* the INR is forced to optimize all frequency bands simultaneously; high-frequency layers overfit first, so fine textures emerge before the coarse layout has stabilized; the premature detail introduces stripe-like artifacts and drives FID to its worst value. Dropping AWP *(iii)* preserves the coarse-to-fine dynamic but does not constrain weights to remain on the manifold defined by the robust anchor, introducing neural artifacts; this allows the unconstrained result to align more closely with the caption, increasing CLIPSIM at the expense of realism (FID ↑). Similarly, eliminating the orthogonal Procrustes projection *(iv)* pushes the optimization toward the raw text embedding, i.e. slightly outside the image sub-manifold: CLIP rewards the closer alignment (CLIPSIM ↑), but the outputs become noticeably busier, with sharper outlines, cluttered details, and occasional duplicated elements. Finally, disabling the blending loss *(v)* stops the optimizer from referencing real-photo statistics; colors turn harsher and small objects

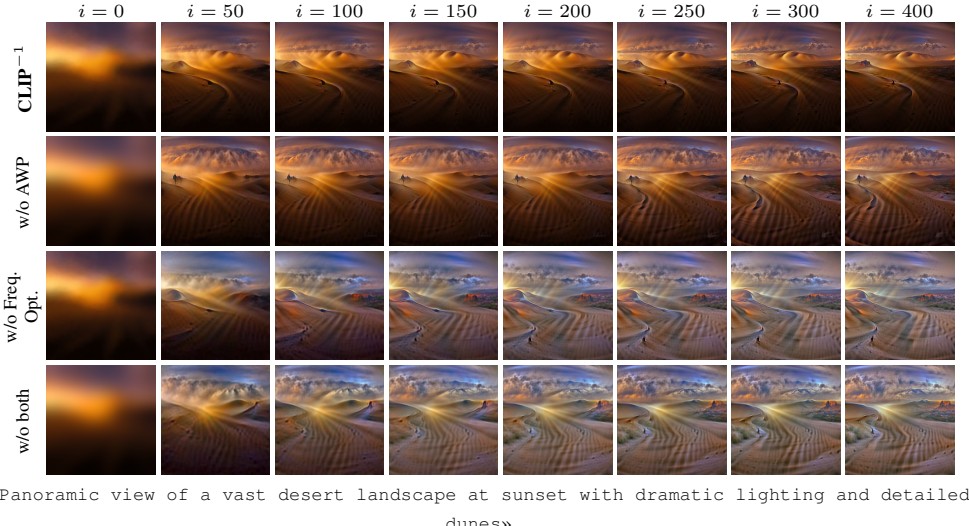

Figure 6: **Qualitative ablation.** Text-to-image synthesis of a desert landscape over 400 iterations, comparing $CLIP^{-1}$ with ablations: without AWP, without frequency-based optimization, and without both.

appear in duplicate, which degrades FID yet inflates CLIPSIM as the model over-expresses caption tokens. In the full model *(i)*, frequency scheduling suppresses glitches, AWP keeps the solution near a stable anchor, and both Procrustes and the blending loss guide the search along the natural-image manifold, trading a few points of raw caption similarity for substantially higher visual quality. The qualitative grid mirrors the numbers: *(i)* is the only variant that is simultaneously aesthetic, coherent, and semantically faithful. Figure 6 traces a prompt through 400 inversion steps to show how the most critical components influence the optimization. Two broad patterns emerge. *(i)* Frequency scheduling governs the refinement path: when it is present (top two rows) the image is generated in a coarse-to-fine order where color appears first, then shapes, then texture; without it (rows 3 & 4) high-frequency stripes appear almost immediately and persist. *(ii)* AWP mitigates cumulative drift: when it is present (rows 1 & 3) the global scene layout stays stable throughout optimization, whereas its absence (rows 2 and 4) lets distortions and noise grow with every iteration. The variant lacking both safeguards shows the combined failure modes, underscoring their complementary roles.

**Ablation for number of inversion steps.** Table 3c shows instead the performance varying the number of inversion steps. Similar to the recurrence of flow-based model, we also have a recurrence when inverting. Here we show that even tough we increase the steps to 400 our models does not overfit likes the other; also, if we decrease the inversion to 40 steps for faster synthesis, the performance drops but still higher than current competing inversion methods.

## 5 CONCLUSION

We present $CLIP^{-1}$, an inversion-based approach that uses a frozen CLIP image encoder and no pretrained decoder for text-to-image synthesis through implicit neural representations (INRs). Instead of relying on a generative decoder, we show that CLIP, combined with a frequency-aware INR and a lightweight alignment step, can guide image synthesis directly from text prompts. Our aim is not to match state-of-the-art generators, but to highlight an underexplored capability: a frozen discriminative model can still produce coherent, semantically aligned images without any additional training. The same setup also handles zero-shot reconstruction, controlled edits, and neural style transfer, and offers an interpretability tool: CLIP inversion reveals how the encoder responds to negations and out-of-distribution prompts, helping expose biases or misalignments before they propagate into larger multimodal systems. These results suggest new directions for repurposing pretrained models and broader implications for robustness and interpretability. Current limitations reflect the nature of CLIP inversion: fine details remain under-constrained and may introduce local artifacts, since all regularization comes from the INR and the CLIP-based losses. A lightweight natural-image prior or a projection step onto the manifold could further improve fidelity.

## ETHICS, AND REPRODUCIBILITY

**Ethics.** We assert that this work does not raise identifiable ethical concerns or foreseeable negative societal consequences. Rather, our contributions point toward improving the explainability of classifier models and their hidden biases, and toward future extensions enabling image generation on commodity hardware and controllable image editing. Our generator, however, is guided by a frozen CLIP model and therefore inherits its known societal biases, potential misuse risks (Kazemi et al., 2024), and conceptual limits. To mitigate these issues, we performed preliminary experiments with Safe-CLIP (Poppi et al., 2024), which successfully blocked harmful content, providing encouraging evidence that safety-aware integrations can be effective.

**Reproducibility.** For reproducibility, we carefully document our full pipeline in (§A.4), along with the complete AWP algoritm (§A.3), providing a step-by-step description of the inversion process. Detailed implementation settings, including all hyperparameters used in our experiments, are further reported in (§A.5).

**Acknowledgment.** This work was supported by projects PNRR MUR PE0000013-FAIR under the MUR National Recovery and Resilience Plan funded by the European Union - NextGenerationEU, PRIN 2022 project 20227YET9B "AdVVent" CUP code B53D23012830006, and MUR FIS2 grant n. FIS-2023-00942 "NEXUS" (cup B53C25001030001). It was also partially supported by Sapienza research projects D2QNeT and BEAT (Better dEep leArning securiTy) — bandi per la ricerca di Ateneo 2024—and Seed of ERC grant "MINT.AI". Computing was supported by CINECA cluster under project Ge-Di HP10CRPUVC, RDM HP10C7YYL2 and the Sapienza Computer Science Department cluster. The authors would like to thank **Andrea Salinetti** for his initial work and bachelor's thesis on CLIP inversion.

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

# A    APPENDIX

This appendix expands on key aspects of our work by providing additional technical details and extended qualitative results. It is organized into eight sections, each addressing a specific area that complements the main paper. Section Section A.1 examines the generative capability of $CLIP^{-1}$, highlighting its ability to synthesize diverse yet semantically consistent images. Section Section A.2 analizes runtime and resource efficiency, comparing memory usage against prior baselines. Section A.3 offers a detailed explanation of the adversarial weight perturbation (AWP) training procedure, including how the perturbation is computed and integrated into the overall training pipeline. Section A.4 outlines the complete algorithm used for the text-to-image task. Section A.5 describes the implementation setup, including model configurations, training parameters, and data preprocessing choices. Section A.10 presents the best and worst 500 generations from MS-COCO prompts, the same samples used in the paper's evaluation. Section A.6 provides qualitative examples generated with different pre-trained ViT-B/32 models, illustrating the variability introduced by different backbone initializations. Finally, Section A.7 includes additional qualitative results from the ablation study, offering visual comparisons that highlight the contributions of individual components. This appendix is intended to support reproducibility and provide a deeper insight into our methodology and experimental findings.

## A.1    GENERATIVE CAPABILITY AND INHERENT STOCHASTICITY

While our full pipeline leverages an informative initialization to improve convergence and quality, the generative capability of $CLIP^{-1}$ does not depend on it. The inversion framework can synthesize images directly from a randomly initialized INR, which effectively acts as a noise input.

In generative models (GMs), GANs sample $z \sim \mathcal{U}[0, 1]$ and diffusion models $z \sim \mathcal{N}[0, 1]$. In our case, fixing the FINER INR's inductive bias, each weight $W_i$ is drawn from $\mathcal{U}\left[-\sqrt{6/n}, \sqrt{6/n}\right]$ and each bias $b_i \sim \mathcal{U}[-p, p]$, where $n$ is the neuron input count, $p$ the initial high-frequency capacity.

This defines a tractable high-dimensional prior ($d \gg k \times k$) over the INR weight space, refined by inversion on the text prompt. Different $z$ produce distinct optimizations and outputs. This is novel in our setting but similar to the multivariate Gaussian in DDPM with $d = k \times k$ or sampling in GAN where $d < k \times k$. Thus, our noise sources are tractable and sufficient for diversity.

To further assess generative behavior, we fixed a text prompt and generated multiple samples. We measured (i) pixel-space variance across runs and (ii) the mean $\pm$ std of CLIP similarity between each generated image and the prompt. Results in Table 4 indicate that $CLIP^{-1}$ achieves substantially higher pixel-space variance than DAS while maintaining comparable CLIP alignment. This confirms that our method produces semantically consistent yet visually diverse outputs. The per-channel variances closely match those of natural images, further validating the realism of the diversity.

Table 4: **Intra-prompt stochasticity**. $CLIP^{-1}$ achieves higher pixel variance while keeping semantic alignment stable.

| Metric | $\mathbf{CLIP^{-1}}$ | DAS |
|---|---|---|
| Variance $\ell_2$ in the pixel space | **0.0447** | 0.0060 |
| Variance $\ell_2$ in the pixel space (per channel) | **[0.0432, 0.0442, 0.0467]** | [0.0052, 0.0070, 0.0057] |
| Alignment to prompt (mean $\pm$ std) | **0.3151 $\pm$ 0.0058** | 0.3033 $\pm$ 0.0078 |

## A.2 Runtime and Resource Analysis

We report the resource usage in Table 6, compared to DAS. The higher VRAM of full $\text{CLIP}^{-1}$ is due to training the Init INR on-the-fly, an overhead removed when cached INRs are reused.

Table 5: **Hardware usage comparison** Average inference time and peak VRAM. (NVIDIA RTX A6000)

| Model | Time (s) | Peak VRAM |
| --- | --- | --- |
| DAS Ensemble | 44.88 | 5.3 GB |
| **$\text{CLIP}^{-1}$ (cached INR, 400 steps)** | **22.88** | **3.2 GB** |
| **$\text{CLIP}^{-1}$ (on-the-fly, 400 steps)** | 29.19 | 4.3 GB |

To evaluate whether the pipeline can run on a mid-range consumer GPU, we tested it on an RTX 4060 (8 GB VRAM). We use 40 optimization steps, since Table 3c shows that this setting does not significantly affect image quality. Compared to GLIDE and LDM-KL-8, $\text{CLIP}^{-1}$ is noticeably more efficient: GLIDE requires more time and memory, while LDM-KL-8 goes out of memory on the same hardware. $\text{CLIP}^{-1}$ therefore remains the method in this comparison that runs most comfortably on lower-end GPUs.

Table 6: **Hardware usage comparison on a consumer GPU** Average inference time and peak VRAM. (NVIDIA RTX 4060, 8GB VRAM)

| Model | Time | Peak VRAM |
| --- | --- | --- |
| DAS Ensemble | 1m 26s | 5.3 GB |
| LDM-KL-8 | - | OOM |
| GLIDE (Text2Img) | 19.2s | 3.7 GB |
| GLIDE (CLIP-Guided) | 9.8s | 3.9 GB |
| **$\text{CLIP}^{-1}$ (40 steps)** | **7.8s** | **3.2 GB** |

## A.3 Further details on the AWP algorithm

We provide a detailed breakdown of the Adversarial Weight Perturbation (AWP) procedure and its integration into the INR training loop. Algorithm 1 outlines the core AWP mechanism: given the weights of the INR model $\phi$ and the input coordinates $(i, j)$ and a temporary clone $\widehat{\phi}$ is optimized to maximize the negative structural similarity index (SSIM) loss between the predicted output and a blurred version of the ground-truth image $\mathbf{x}$. The resulting adversarial perturbation $\Delta\phi$ is computed, normalized, and applied to the original weights $\phi$ to obtain the perturbed weights $\phi_{\text{adv}}$.

---

**Algorithm 1:** Adversarial Weight Perturbation (COMPUTE_AWP)

1 **Inputs:** INR model weights $\phi$, input coordinates $(i, j)$, target image $\mathbf{x}$
2 $\widehat{\phi} \leftarrow \text{clone}(\phi)$             `// proxy model initialization`
3 $\mathcal{L}_{\text{awp}} \leftarrow -\mathcal{L}_{\text{SSIM}}\big(f_{\widehat{\phi}}((i, j)), \text{blur}(\mathbf{x})\big)$        `// maximize the loss`
4 Optimize $\widehat{\phi}$ w.r.t. $\mathcal{L}_{\text{awp}}$
5 $\Delta\phi \leftarrow \widehat{\phi} - \phi$             `// compute perturbation`
6 $\Delta\phi \leftarrow \gamma \cdot \frac{\|\phi\|}{\|\Delta\phi\|+\epsilon} \cdot \Delta\phi$       `// normalize and scale perturbation`
7 **Return:** $\Delta\phi$

---

Algorithm 2 illustrates the incorporation of AWP into INR training. At each iteration, adversarial perturbations are computed using Algorithm 1 and then applied to the network. The overall training loss is a weighted combination of mean squared error (MSE), SSIM, and $\ell_1$ loss. This adversarial training scheme improves the robustness and generalization of the INR by encouraging consistency under weight-level perturbations, which are the gradients received by inverting CLIP when generating.

---

**Algorithm 2:** INR Training with AWP

---

1  **Inputs:** target image $\mathbf{x}$, initial weights $\phi_0$, input coordinates $(i, j)$
2  **Hyperparameters:** learning rate $\eta$, perturbation scale $\gamma$, iterations $N$
3  **for** $k = 1, \dots, N$ **do**
4     $\Delta\phi \leftarrow \text{COMPUTE\_AWP}(\phi_k, (i, j), \mathbf{x})$
5     $\phi_{\text{adv}} \leftarrow \phi_k + \Delta\phi$                                       `// apply perturbation`
6     $f_{\phi_{\text{adv}}} \leftarrow$ model with weights $\phi_{\text{adv}}$
7     $\hat{\mathbf{x}} \leftarrow f_{\phi_{\text{adv}}}((i, j))$
8     $\mathcal{L} \leftarrow \alpha_1 \mathcal{L}_{\text{MSE}}(\hat{\mathbf{x}}, \text{blur}(\mathbf{x})) + \alpha_2 \mathcal{L}_{\text{SSIM}}(\hat{\mathbf{x}}, \text{blur}(\mathbf{x})) + \alpha_3 \mathcal{L}_{\text{L1}}(\hat{\mathbf{x}} - \text{blur}(\mathbf{x}))$
9     Update $\phi_{\text{adv}}$ via optimizer step minimizing $\mathcal{L}$
10    $\phi_{k+1} \leftarrow \phi_{\text{adv}} - \Delta\phi$          `// restore original weights for next iteration`
11  **Return:** trained weights $\phi_N$

---

## A.4   TEXT-TO-IMAGE FULL PIPELINE

We optimize an implicit neural representation (INR) to synthesize an image that aligns with a given text prompt using CLIP. The procedure includes text and image retrieval, feature alignment, and iterative gradient-based optimization. Procrustes alignment and natural image constraints are enabled. Algorithm 3 shows the detailed pipeline.

---

**Algorithm 3:** Text-to-Image Synthesis

---

1  **Input:** Text prompt $\mathbf{y}$,
2  **Output:** Synthesized image $\mathbf{x} = f_{\phi^N}((i, j))$
3

   `/* Input pre-processing and alignment`                            `*/`
4  Encode the input prompt $\mathbf{e}_t = \boldsymbol{\theta}_T(\mathbf{y})$
5  Select top-$k$ matches to $\mathbf{e}_t$ in $\mathcal{D}$ to build $\mathbf{E}_T, \mathbf{E}_I \in \mathbb{R}^{d \times k}$
6  Compute on the fly the orthogonal Procrustes rotation matrix
    $R = \min_{\mathbf{R}} \|\mathbf{R}\mathbf{E}_T - \mathbf{E}_I\|_F$     s.t. $\mathbf{R}^\top \mathbf{R} = \mathbf{I}$
7  Project $\mathbf{e}_t$ to visual domain: $\mathbf{e}_{t2i} = \mathbf{R}\boldsymbol{\theta}_T(\mathbf{y})$
8

   `/* Retrieve Initialization:`                                               `*/`
9  From dataset $\mathcal{D}$, retrieve image embedding $\boldsymbol{\theta}_I(\hat{\mathbf{x}})$ of $\hat{\mathbf{x}}$ with caption closest to $\mathbf{e}_{t2i}$
10  Initialize INR weights $\phi_0 \leftarrow \text{INIT\_INR\_AWP}(\hat{\mathbf{x}})$—see Algorithm 2
11

   `/* Natural Image Constraints:`                                        `*/`
12  Retrieve top-$k$ natural images $\{\mathbf{x}_j^\star\}$ near $\mathbf{e}_t$ in CLIP space
13  Encode them to features $\{\mathbf{e}_{img,j}^\star\}$
14  Store the similarity to the input prompt $w_j = \text{CLIPSIM}(\mathbf{x}_j^\star, \mathbf{e}_t)$
15  Compute weighted average: $\mathbf{e}_{img}^\star = \sum_j w_j \mathbf{e}_{img,j}^\star$ where $w_j$ are normalized w/ softmax
16

   `/* Optimizer Setup:`                                                 `*/`
17  Initialize layer-wise optimizers with Gaussian learning rates (peak is $\gamma$) over INR depth.
18  Schedule shifting of Gaussian center every $k$ steps—see Fig. 7
19
20  **for** $i = 1$ *to* $T$ **do**
21     **if** *learning rate schedule triggers* **then**
22         Shift Gaussian center layer
23     Encode via CLIP the augmentations of the rendered INR:

$$\mathbf{e}_i^\star = \frac{1}{n} \sum_{k=1}^n \boldsymbol{\theta}_I \big( \text{augment}(f_{\phi^k}) \big)$$

24     Compute total loss: $\mathcal{L}(\mathbf{e}_i^\star, \mathbf{e}_{t2i}) + \beta \mathcal{L}_{blend}(\mathbf{e}_i^\star, \mathbf{e}_{img}^\star)$
25     Update $\phi$:

$$\phi_n = \phi_{n-1} - \nabla_\phi \Big[ \mathcal{L}(\mathbf{e}_i^\star, \mathbf{e}_{t2i}) + \beta \mathcal{L}_{blend}(\mathbf{e}_i^\star, \mathbf{e}_{img}^\star) \Big]$$

26  **Return:** Final image $\mathbf{x} = f_{\phi^N}((i, j))$

---

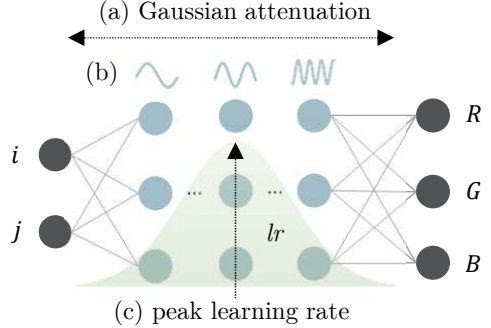

Figure 7: **Gaussian Scheduling.** Each layer represents a frequency interval *(b)* for $f_\phi$. The learning rate is centered on a specific layer and gradually shifted *(c)*, decreasing with a Gaussian attenuation across neighboring layers *(a)*.

## A.5 IMPLEMENTATION DETAILS

### A.5.1 INR PARAMETERS AND INITIALIZATION

We initialize our implicit neural representations (INRs) with `in_features = 2` and `out_features = 3`, using five hidden layers of 256 units each. Sinusoidal parameterization is applied with `first_omega = 25` and `hidden_omega = 25` to enable high-frequency signal modeling. Training is performed using the Adam optimizer with a learning rate of $1 \times 10^{-4}$, and a cosine annealing schedule via `torch.optim.lr_scheduler.CosineAnnealingWarmRestarts` with a restart period of 100 iterations.

We apply Adversarial Weight Perturbation (AWP) using a proxy optimizer with the same learning rate and a perturbation strength of $\alpha = 0.01$. The ground-truth image $\mathbf{x}$ is preprocessed using a Gaussian blur with `kernel_size = 101` and $\sigma \in (10.0, 20.0)$ to provide a smoother supervision signal.

The training loss combines mean squared error (MSE), structural similarity (SSIM), and $\ell_1$ reconstruction loss, weighted, respectively, by $\alpha_1 = 0.85$, $\alpha_2 = 0.25$, and $\alpha_3 = 0.25$.

### A.5.2 TEXT-TO-IMAGE INVERSION PARAMETERS

Our method is built upon a *ViT-B/32* backbone initialized with the default OpenAI weights. We perform 400 inversion steps using the *AdamW* optimizer (without *AMSGrad*) and a learning rate of $2 \times 10^{-4}$. During INR optimization, we employ a Gaussian scheduling strategy focused on layers $[0, 1, 2]$, with gradient norm clipping thresholds set to $[1.0, 0.5, 0.2]$ respectively. This schedule is refreshed every 70 iterations to preserve both stability and optimization efficiency over time.

The loss function incorporates hyperparameters $\beta = 0.5$ and $k = 8$, balancing the trade-offs between reconstruction fidelity and robustness. To promote generalization, we apply data augmentation by generating 32 variations per input sample. For spatial alignment, we use Orthogonal Procrustes analysis over the nearest $p = 256$ elements. Following Stable Diffusion (Jagielski et al., 2023), we guide the CLIP inversion process by appending auxiliary textual prompts that explicitly describe desired image characteristics. This strategy improves the fidelity and perceptual quality of the generated outputs.

## A.6 QUALITATIVE SAMPLES UNDER DIFFERENT CLIP MODELS

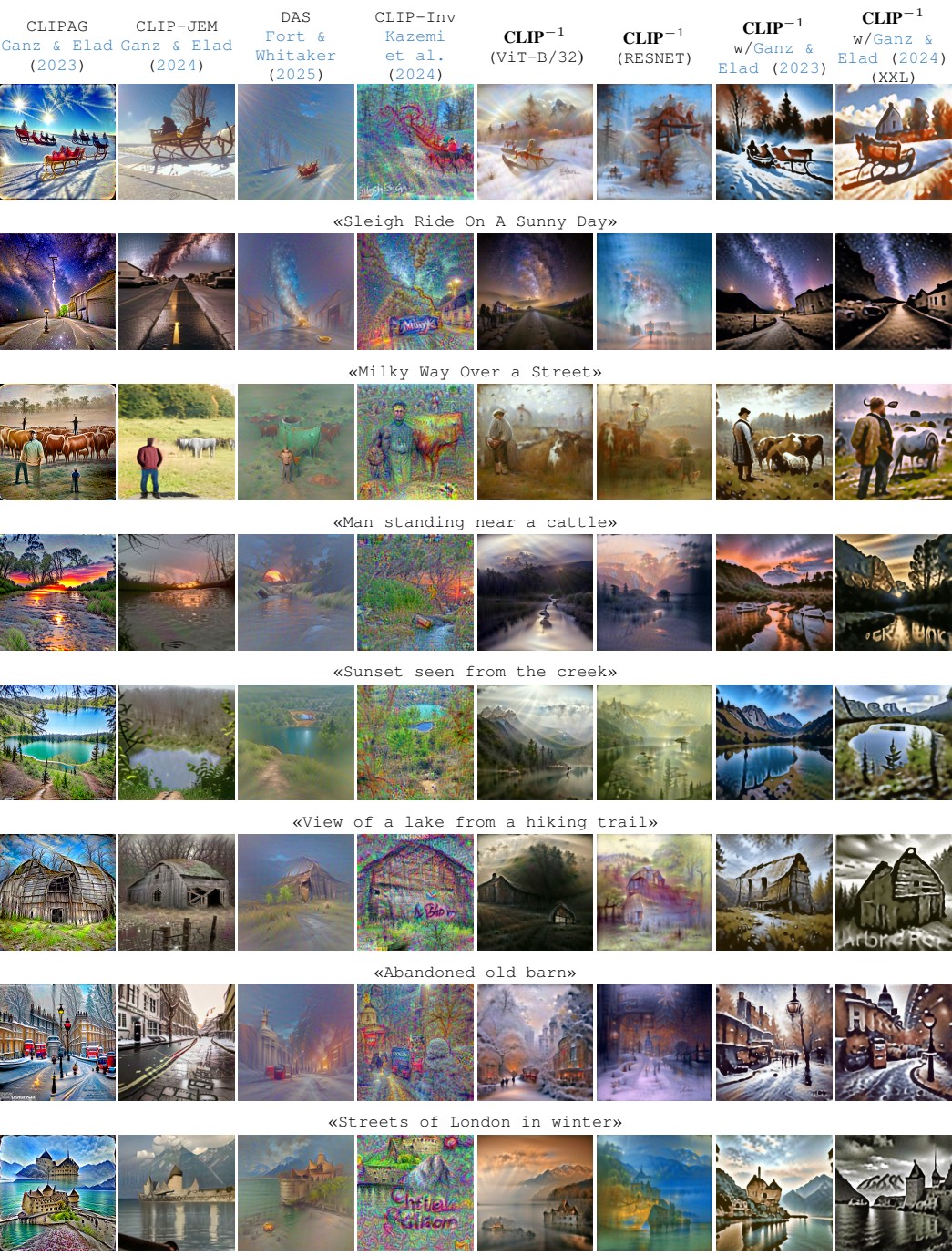

Figure 8: **Qualitative comparison** of additional samples extending Figure 4 in the main paper.

## A.7 QUALITATIVE SAMPLES OF THE ABLATION STUDY

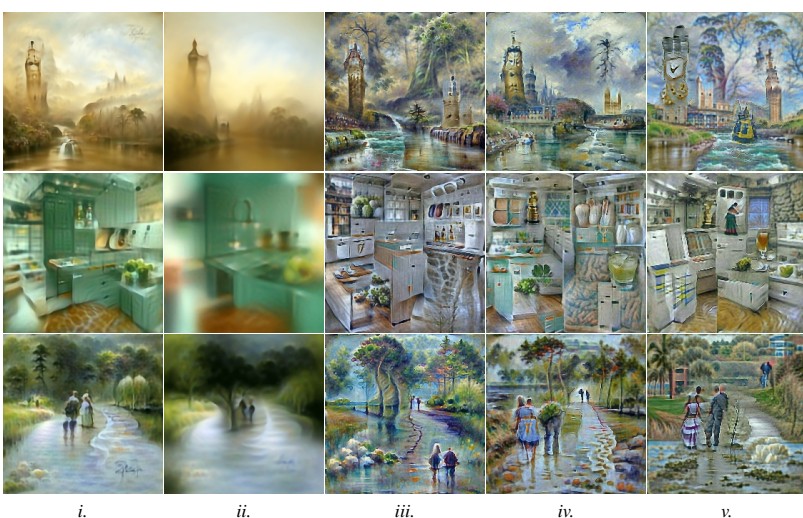

*i.*    *ii.*    *iii.*    *iv.*    *v.*

Figure 9: **Quantitative ablation study.** (a) Results on 1,000 MS-COCO captions. (b) Samples for each case within the same prompt; columns show: – *i.* full model – *ii.* frequency scheduling – *iii.* AWP – *iv.* F.O. + Procrustes – *v.* F.O. + blending loss.

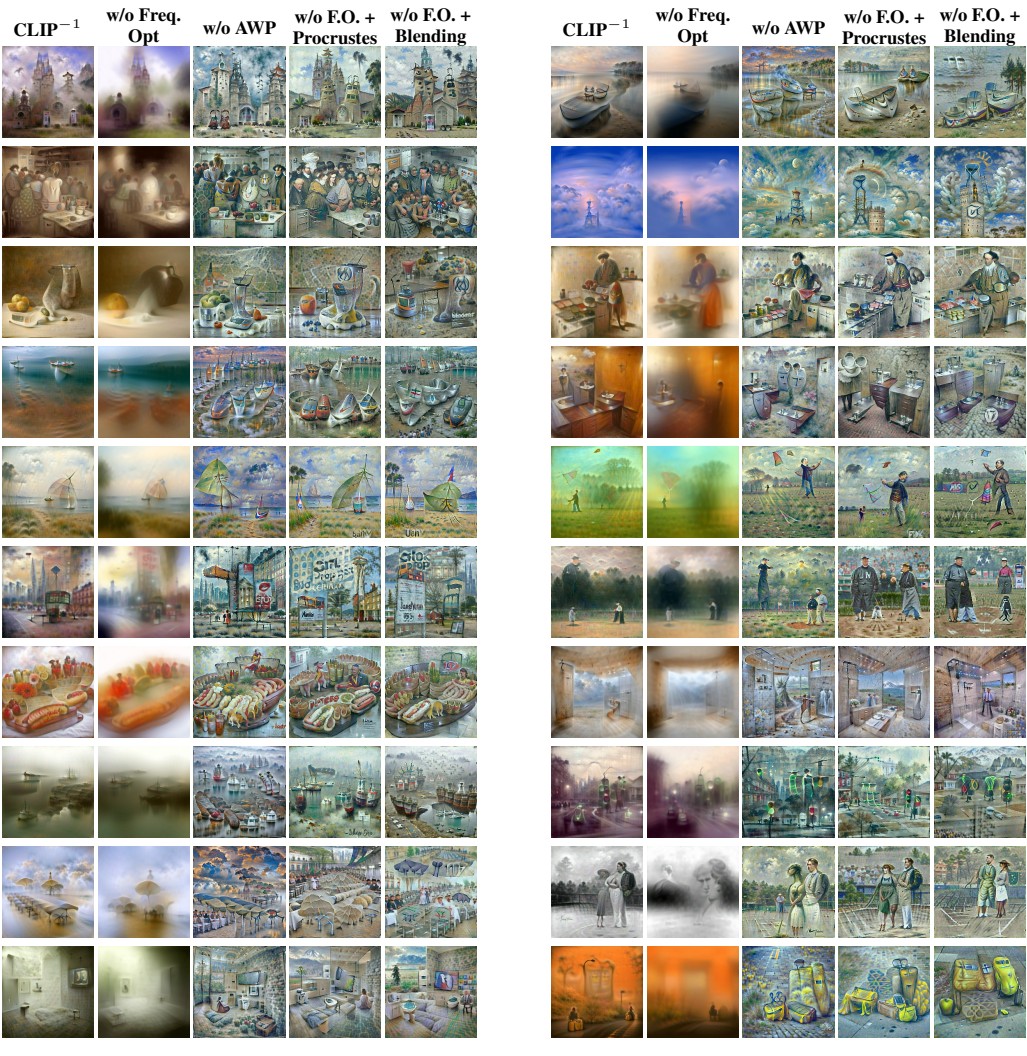

**Left:** CLIP$^{-1}$ Ablation Rows 1-10          **Right:** CLIP$^{-1}$ Ablation Rows 11-20

Figure 10: **Ablation Study** additional samples of the ablation study shown in Figure 6 of the main paper.

## A.8 HIGHER RESOLUTIONS

Images are synthesized at $448 \times 448$, which is $2\times$ the original CLIP input resolution. Below we compare the original image, its upscaled version, and the corresponding native synthesis. Generating at higher resolution also induces the model to introduce additional content, as seen in the monastery-on-a-cliff image, where a second building appears in the reconstruction.

Original (upscaled)  Native 448x448

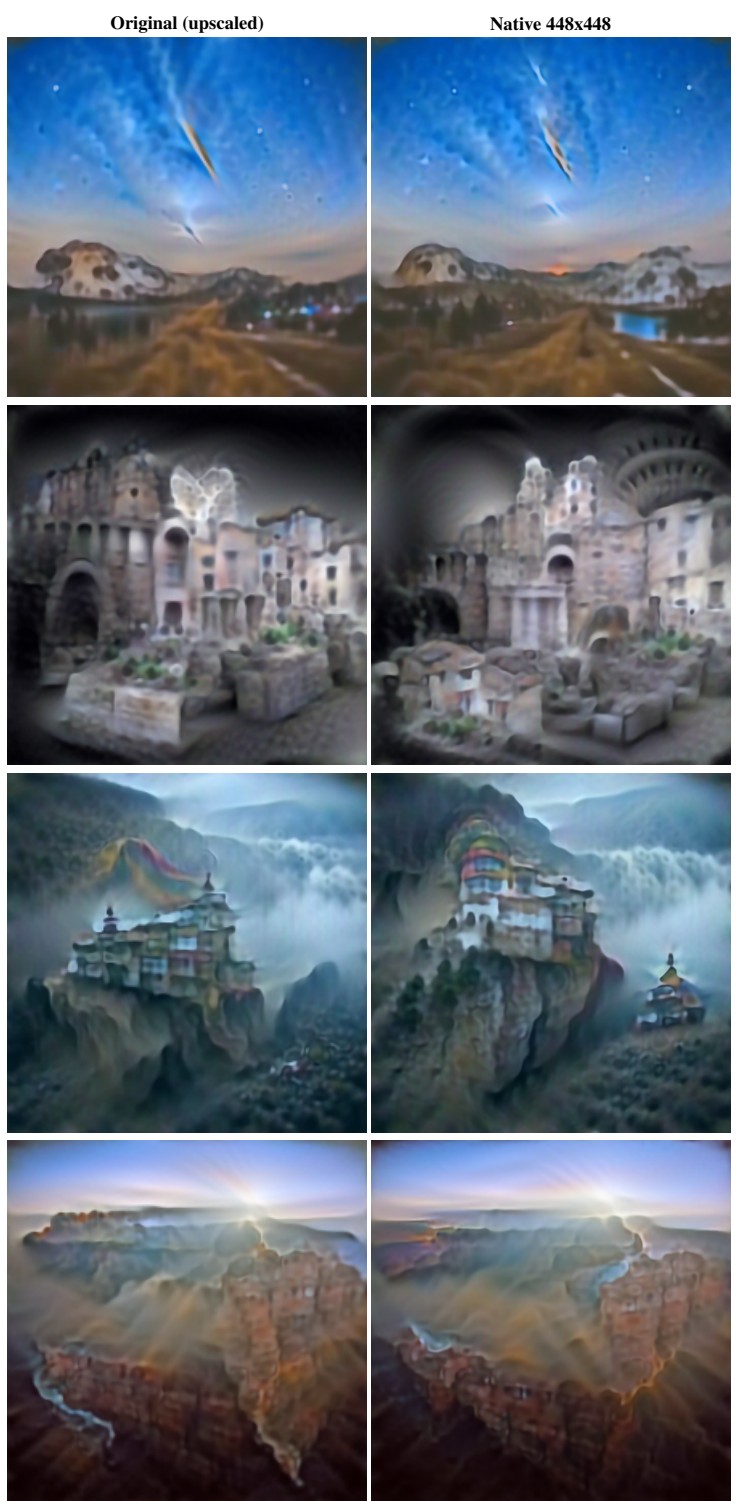

Figure 11: **Native text-to-image at higher resolution.** Left: images generated at $224 \times 224$ and upscaled. Right: images generated natively at $448 \times 448$.

## A.9 ROBUSTNESS TO WRONG AWP INITIALIZATIONS

We assess robustness by introducing three types of wrong AWP Initializations that may happens when implementing Section 3.2.

We fixed an initial robust AWP seed matching a prompt and then we invert the correct prompt. This corresponds to the generation with the correct retrieved initialization related to a lake. Then we do the following: we kept fixed the initialization of the correct prompt, but invert four different prompts uncorrelated with current initialization. The uncorrelated prompts are:

- (a) Some oranges are stacked up in a bowl
- (b) A pastry store with cupcakes on display
- (c) Tibetan monastery on a cliff
- (d) A group of people sitting around a table

We show the result above for three different starting seeds in Fig. 12

The first $(i)$ is a spacious and uncluttered image, such as a lake, used when the target scene is highly cluttered. The second $(ii)$ corresponds to a seed built from a complex out-of-distribution concept. The third $(iii)$ is a seed whose colors and overall appearance do not match the concepts described in the other prompts.

The results show that the semantics remain largely unchanged, which indicates robustness. The seed mainly affects the spatial arrangement of objects and, more strongly, the overall color palette. Even when the palettes differ, the coloring scheme remains semantically consistent across the objects in the scene. In case $(iii)$, however, the initialization departs more from what is expected, due to the nature of the adversarial seed itself (hard edges, city-like structures, neon lights), and this leads to more visible artifacts.

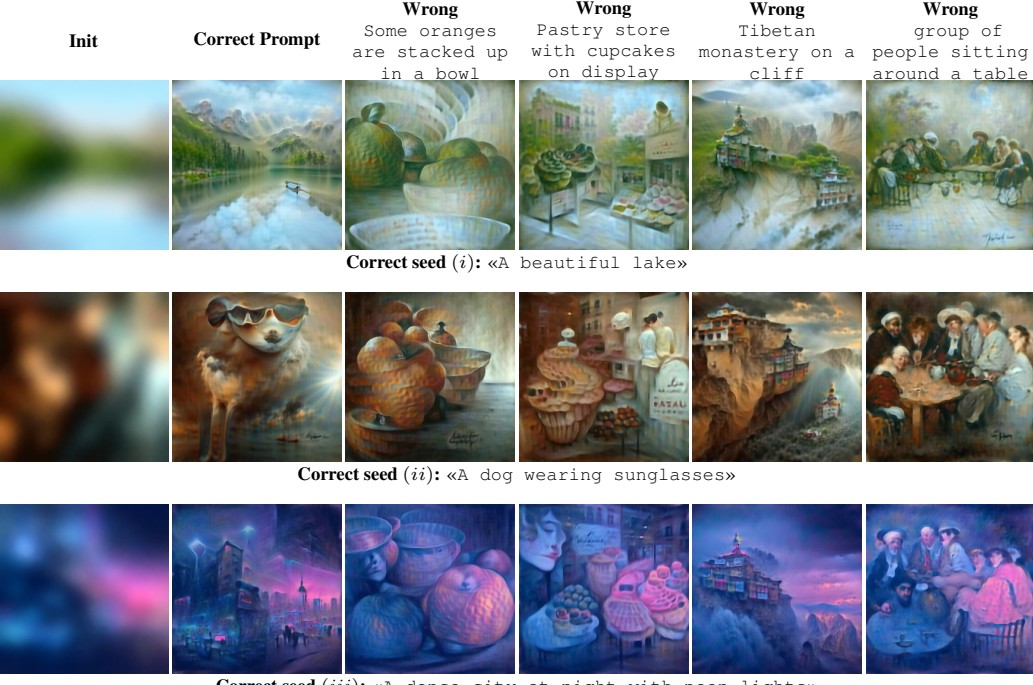

Figure 12: **Behavior with the wrong initialization** We show how generation performs when the initialization does not match the inverted prompt.

## A.10  BEST VS. WORST MS-COCO GENERATIONS USING CLIP SCORE AS METRIC

To qualitatively assess model's performance, this section presents the 500 highest and 500 lowest scoring generations based on CLIP Score, using prompts from the MS-COCO captions dataset. Fig. 13 shows the best generations that achieved CLIP scores between 45.5 and 53.9 (mean: 47.2). Fig. 14 shows the worst, ranging from 23.0 to 32.2 (mean: 30.6). These examples illustrate the range of output quality, from strong semantic alignment to notable failure cases.

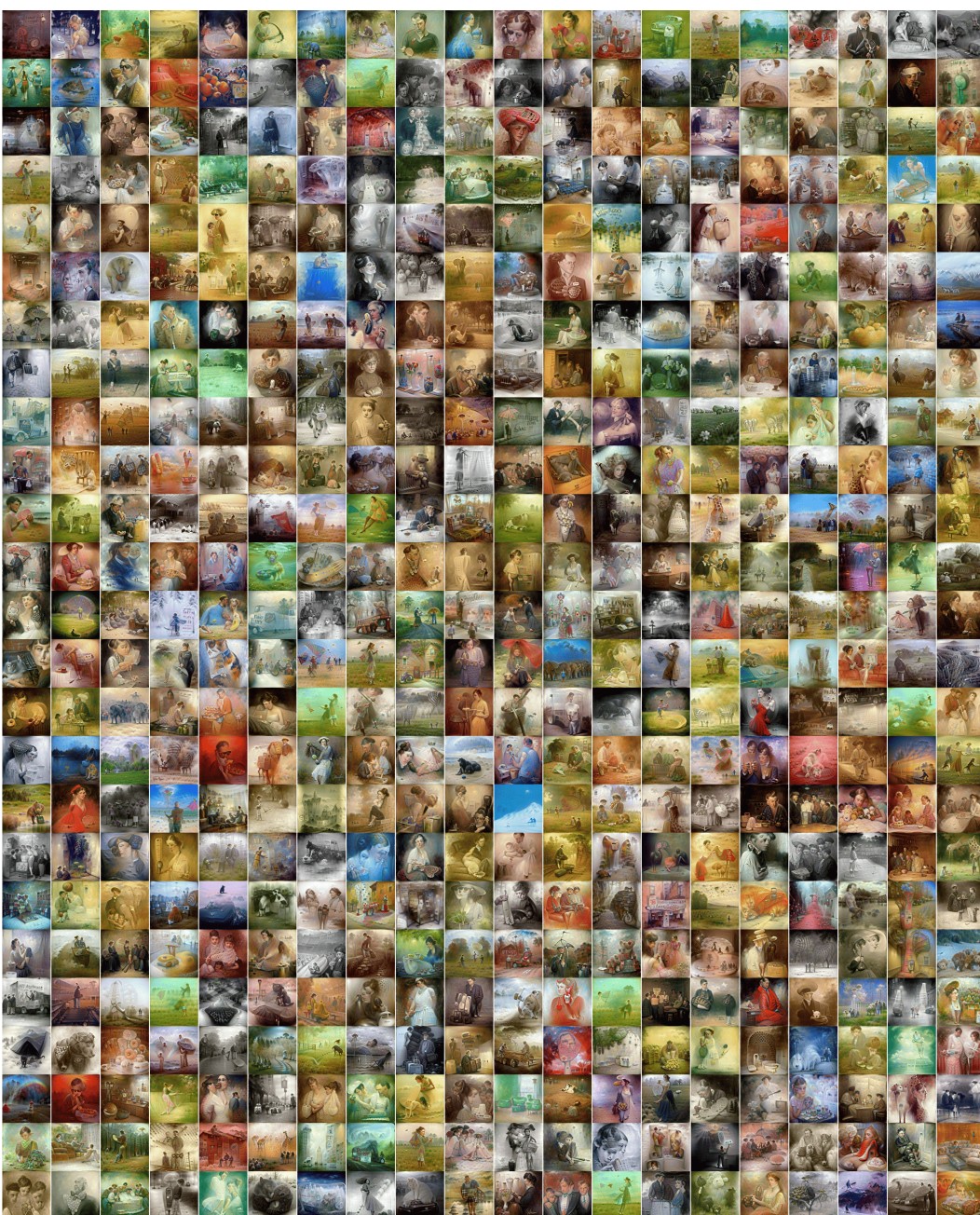

Figure 13: **Best generations on MS-COCO prompts**. CLIP Score ranging from 45.5 to 53.9 (mean: 47.2)

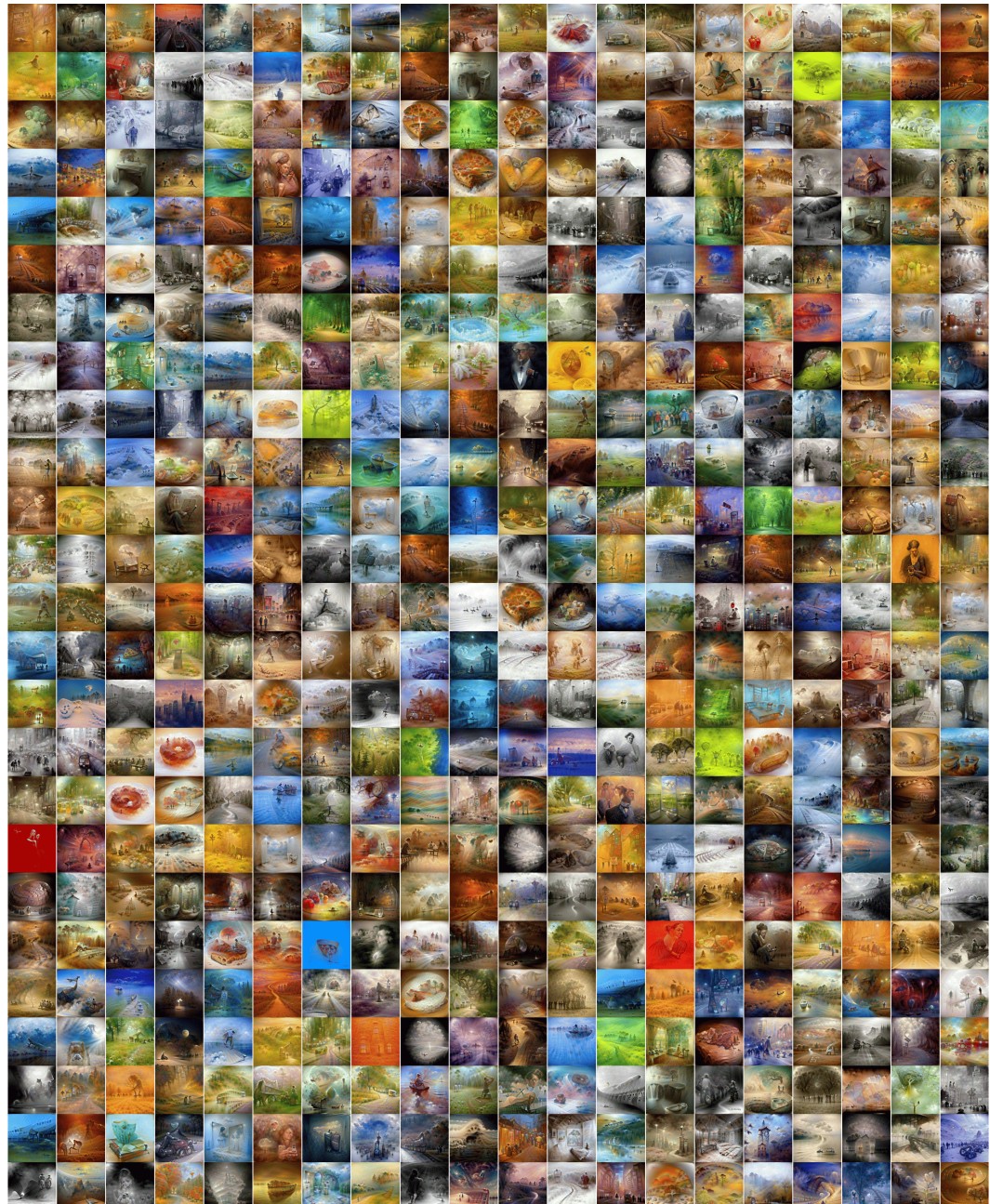

Figure 14: **Worst generations on MS-COCO prompts**. CLIP Score ranging from 23.0 to 32.2 (mean: 30.6)

## A.11 LIMITATIONS

Our method's quality is inherently tied to the representations encoded in CLIP: performance may degrade for prompts that are abstract, rare in CLIP's training data, or require fine compositional detail (e.g., "a woman wearing planet-shaped earrings"). These challenges are intrinsic to classifier-inversion approaches, and we plan to address them more systematically in future work.

## A.12 LLM USAGE

Large language models were used exclusively for text polishing and minor exposition refinements. All substantive research content, methodology, and scientific conclusions were developed entirely by the authors.

