# OpenReview forum: "Implicit Inversion turns CLIP into a Decoder"
_ICLR.cc/2026/Conference — ICLR 2026 Poster_

### Official Review · Reviewer_fJkf · 2025-10-22

**Soundness:** 3
**Presentation:** 4
**Contribution:** 4
**Rating:** 6
**Confidence:** 4

**Summary:**

This paper introduces $CLIP^{-1}$, a novel framework for text-to-image synthesis that demonstrates the latent generative capabilities of a frozen, pre-trained discriminative model (CLIP) . The central claim is that a dedicated, trained generative decoder is not strictly necessary. Instead, the authors reframe image synthesis as an inversion problem: finding an image that, when encoded by CLIP's image encoder, matches a target text embedding .The paper's core contribution is a set of techniques to solve this notoriously ill-posed problem, which typically yields unstable, non-naturalistic images (adversarial examples) . The authors' solution avoids direct pixel optimization and instead optimizes the weights of a frequency-aware Implicit Neural Representation (INR), which inherently enforces smoothness and a structured, coarse-to-fine generation process .To further stabilize this inversion, the $CLIP^{-1}$ pipeline introduces three key components:An adversarially robust initialization using Adversarial Weight Perturbation (AWP) on a blurred seed image, which anchors the optimization in a stable "flat" region of the loss landscape .A lightweight, geometric solution to the modality gap using Orthogonal Procrustes Analysis to project the target text embedding onto the image embedding manifold, providing a more stable target .A blending loss that acts as a natural image prior, regularizing the output by encouraging it to remain close to the statistics of real image embeddings .Empirically, $CLIP^{-1}$ substantially outperforms other training-free inversion methods like DAS, achieving a Fréchet Inception Distance (FID) that is less than half of its competitor . The framework also demonstrates zero-shot versatility on downstream tasks, including image reconstruction, controlled text-guided editing, and neural style transfer.

**Strengths:**

1) The paper's core idea, unlocking the generative potential hidden within a frozen discriminative model, is highly original and significant. It moves beyond simply using CLIP as an encoder and shows it can be "run in reverse" as a generator, challenging the conventional wisdom that separates discriminative and generative models.

2) The key innovation is reframing the inversion problem from pixel space to the weight space of an Implicit Neural Representation (INR). This choice enforces smoothness, helps in preventing high-frequency artifacts and enables a structured coarse-to-fine optimization by leveraging the frequency-stratified architecture of the FINER network.

3) The paper introduces simple, effective, and computationally cheap solutions to complex problems. I like the use of Orthogonal Procrustes Analysis to geometrically correct the modality gap, as it requires no training and is demonstrably effective.

4) The combination of Adversarial Weight Perturbation (AWP) for initialization and the blending loss creates a more stable optimization process. These components successfully anchor the search and prevent it from converging to the "adversarial examples" (perceptually meaningless noise) that have plagued previous inversion attempts.
5) The ablation study is thorough and provides a clear justification for each component of the $CLIP^{-1}$ pipeline.

**Weaknesses:**

1) The paper's primary weakness is its discussion of the computational cost. The method is "training-free", but it shifts the entire computational load to inference time, which requires an iterative optimization process (400 steps). The runtime analysis in the appendix compares $CLIP^{-1}$ only to DAS, another inversion-based method. The main quantitative comparison compares quality (FID/IS) against SOTA generative models like LDM and GLIDE but critically omits a comparison of inference time and VRAM usage. A single forward pass through a diffusion decoder is likely much faster than the 1+ minute optimization required by $CLIP^{-1}$. To fairly evaluate this as a generative "decoder" alternative, this trade-off must be made explicit.
2) As the authors rightly acknowledge, a significant gap in photorealism and fine-detail fidelity remains when compared to SOTA diffusion models. The generated images, while coherent, often have a "painterly" or slightly artificial quality and lack crisp, high-frequency textures. This limits the method's immediate practical application for photorealistic generation.Initialization.
3) The full, best-performing model relies on retrieving a semantically relevant seed image from the LAION dataset to perform its AWP-based initialization. While the out-of-distribution evaluation shows that a "Plain $CLIP^{-1}$" variant (starting from random weights) can still function, the main method's output is still influenced by this seed. This could constrain the creative novelty of the generations, potentially biasing them towards common compositions found in the retrieval database.

**Questions:**

Following up on the main weakness:
1) Could the authors provide a direct comparison of the inference time (time per image on a standard GPU like an A100 or RTX 4090) and peak VRAM usage for $CLIP^{-1}$ against the SOTA generative baselines listed in Table 1 (e.g., LDM-KL-8 or GLIDE)? This would be invaluable for understanding the practical trade-off between the method's training cost and its inference cost.
2) The optimization process is set to 400 steps. Have the authors experimented with this hyperparameter? How does image quality (FID) degrade as the number of steps is reduced? Is there a "sweet spot" that balances quality and speed?
3) This question is about the robustness of the initialization strategy. The paper's AWP-trained seed image is described as a "robust anchor". We are interested in how strongly this anchor constrains the final generation.
- First, the paper shows a "Plain $CLIP^{-1}$" variant that starts from random INR weights. Could the authors provide a qualitative comparison for this variant? Does it simply converge slower, or does it suffer from more significant artifacts, confirming the necessity of the AWP anchor?
- Second, what happens in an adversarial scenario where the retrieved seed image is semantically incorrect (e.g., the prompt is "a photo of a dog" but the retrieved seed is "a photo of a car")? Is the optimization pipeline, guided by the Procrustes-corrected text embedding and the blending loss, strong enough to correct this completely wrong starting point and still generate a "dog"? Or does the robust anchor effectively trap the optimization near the (incorrect) seed, leading to a failed generation?
4) The use of Orthogonal Procrustes is a very clever and simple solution for the modality gap. Did the authors experiment with any alternative methods, such as learning a small, lightweight mapping network (e.g., a simple MLP) to bridge the gap? Was Procrustes chosen primarily for its simplicity and training-free nature, or did it also empirically outperform other simple alternatives?

---

> ### Author Response · Authors · 2025-11-22
> **Answer to Reviewer fJkf (part 1/3)**
>
> We thank `Rev. fJkf` for the strong endorsement of the paper’s originality.
> The reviewer highlights the central idea of revealing the latent generative potential of an off-the-shelf discriminative model. Moreover, we thank the reviewer for recognizing the advantages of the INR-based inversion strategy, as well as the simplicity and effectiveness of components such as Orthogonal Procrustes and AWP initialization. These observations align closely with the strengths noted by `Rev. kSFm`, who emphasizes the same innovations and their complementary roles. `Rev. fJkf` as well appreciates the clarity of the ablations and the coarse-to-fine structure, reflecting feedback consistent with `Rev. JCC3` and `Rev. kSFm`. We address the reviewer's concerns below.
>
> ### W1, Q1, Q2 - Inference time and optimization steps
>
> > The runtime analysis in the appendix compares CLIP^-1 only to DAS, another inversion-based method. The main quantitative comparison compares quality (FID/IS) against SOTA generative models like LDM and GLIDE but critically omits a comparison of inference time and VRAM usage. A single forward pass through a diffusion decoder is likely much faster than the 1+ minute optimization required by CLIP^-1. To fairly evaluate this as a generative "decoder" alternative, this trade-off must be made explicit.
>
> We thank the reviewer for the observation.
> We included a new section in Appendix A.2 that addresses the present concern. We first evaluate the CLIP^-1 inversion method with the DAS ensemble, looking at hardware usage, where our approach significantly reduces the VRAM used and also reduces the inference time. The results are reported in Table 5.
>
> We then further evaluate our inversion method, comparing it with the mentioned generation methods (LDM-KL-8 and GLIDE), using a widely available consumer GPU (RTX 4060) with 8GB of VRAM. **The new Table 6 (also shown below)** reports the comparison results, showing that the proposed approach, evaluated over 40 iterations, generates in 7.8s, significantly reducing the time with respect to the DAS ensemble and also with respect to GLIDE (9.8s).
> We also highlight that our approach is overall the most convenient one in terms of VRAM usage.
>
> | Model | Time | Peak VRAM |
> |-----------------------------|--------|-----------|
> | DAS Ensemble | 1m 26s | 5.3 GB |
> | LDM-KL-8 | - | OOM |
> | GLIDE (Text2Img) | 19.2s | 3.7 GB |
> | GLIDE (CLIP-Guided) | 9.8s | 3.9 GB |
> | **CLIP⁻¹ (40 steps)** | **7.8s** | **3.2 GB** |
>
>
> This clarifies the central trade-off: CLIP$^{-1}$ avoids a learned generative decoder and therefore requires iterative optimization, but in return it provides a decoder-free framework with lower memory requirements, competitive generation times on consumer hardware, and no training cost. The added section explicitly states that CLIP$^{-1}$ should be viewed as a lightweight, optimization-based alternative, not a drop-in replacement, for fast, single-pass generative decoders.
>
> ### W2 - Painterly or slightly artificial quality
>
> > W2) As the authors rightly acknowledge, a significant gap in photorealism and fine-detail fidelity remains when compared to SOTA diffusion models. The generated images, while coherent, often have a "painterly" or slightly artificial quality and lack crisp, high-frequency textures. This limits the method's immediate practical application for photorealistic generation.Initialization.
>
> We thank the reviewer for the opportunity to clarify this point.
> Part of the aesthetic bias comes from our use of the LAION-Aesthetics subset for retrieval: the dataset contains many artistic and stylised images, and the retrieved INRs naturally inherit those visual characteristics. This reliance on the retrieval corpus is noteworthy: since varying the retrieval corpus yields distinct visual characteristics, it might be possible to intentionally steer the look of inversions by choosing a different source dataset.
> Regarding high-frequency details, we'd like to emphasise that optimising them is a deliberate choice. The INRs enable deliberate frequency optimization, resulting in smoother images with less emphasis on higher frequencies.
> We decided to adopt the present configuration as this is a trade-off that prevents overfitting and mitigates the unstable high-frequency artifacts common in pixel-space inversion.

---

> ### Author Response · Authors · 2025-11-22
> **Answer to Reviewer fJkf (part 2/3)**
>
> ### W3) Influence of the retrieved initialization
>
> > This [the init] could constrain the creative novelty of the generations, potentially biasing them towards common compositions found in the retrieval database.
>
> We acknowledge that the retrieved seed can influence the initial point of the inversion and, in turn, the coarse layout of the generation. This follows from the nature of inversion problems, which are intrinsically ill-posed: many images can map to the same CLIP embedding, and the optimization landscape contains multiple valid solutions. An initialization is therefore required to select a stable region of the manifold and to anchor convergence. However, this does not override the prompt's semantic content. When the retrieved seed and the target prompt are misaligned, the optimization consistently converges to the correct meaning.
>
> We support this observation with **a new experiment**. **Appendix A.9 (Figure 12)** demonstrates that even deliberately incorrect initializations still yield the correct semantics. These results, together with the design discussion in Section 3.2, indicate that while the retrieved seed introduces a structural bias, it does not limit semantic or creative expressiveness.
>
> ### Q3) Robustness of the initialization strategy
>
> > Q3) This question is about the robustness of the initialization strategy. The paper's AWP-trained seed image is described as a "robust anchor". We are interested in how strongly this anchor constrains the final generation.
>
> AWP-trained seed constraints the generation to produce images that are more natural and contain fewer artifacts (see Fig. 10 column _w/o AWP_ and also Fig. 9). This does provide some diversity in the spatial locations where objects appear and in the color palette. With that being said, even if starting with a seed that does not perfectly match the prompt we need to invert, the inversion can still generate the mentioned concepts. In fact, the method remains robust even with semantically incorrect seeds. Since retrieved initializations are heavily blurred, the optimization is not trapped by incorrect content: prompts can still produce the intended concept even when the retrieved seed depicts a different object.
>
> > Q3.1) First, the paper shows a "Plain CLIP^-1" variant that starts from random INR weights. Could the authors provide a qualitative comparison for this variant? Does it simply converge more slowly, or does it suffer from more significant artifacts, confirming the necessity of the AWP anchor?
>
> The qualitative comparison between using AWP and not using AWP was presented in the original version, and due to space constraints, we have moved it to a higher resolution in the ablation studies (**Fig. 9, Appendix A.7 with more ablations in Fig. 10**). The `w/o AWP` column shows that **AWP plays a key role**. Quantitative results in Tab. 3(b) of the revised paper (row iii. `w/o AWP`) further confirm its importance for producing more natural images.
>
> The results without AWP suffer from more significant artifacts, confirming the necessity of our proposed AWP anchor.
>
> > Q3.2) Second, what happens in an adversarial scenario where the retrieved seed image is semantically incorrect (e.g., the prompt is "a photo of a dog" but the retrieved seed is "a photo of a car")? Is the optimization pipeline, guided by the Procrustes-corrected text embedding and the blending loss, strong enough to correct this completely wrong starting point and still generate a "dog"? Or does the robust anchor effectively trap the optimization near the (incorrect) seed, leading to a failed generation?
>
> Thank you for raising these interesting experiments. We present qualitative experiments for this scenario **in the new Fig. 12 of Appendix A.9, "Robustness to Wrong AWP Initializations."**
>
> In short, the results show that the semantics remain essentially unchanged regardless of the retrieved seed, indicating robustness. The seed primarily influences the spatial arrangement of objects and, more significantly, the overall color palette. Even when the palettes differ, the coloring scheme remains semantically consistent across the objects in the scene. So, despite wrong initialization, the system is still able to drive the generation towards the input prompt.

---

> > ### Author Response · Authors · 2025-11-22
> > **Answer to Reviewer fJkf (part 3/3)**
> >
> > ### Q4) Choice of Orthogonal Procrustes
> >
> > > Q4) The use of Orthogonal Procrustes is a very clever and simple solution for the modality gap. Did the authors experiment with any alternative methods, such as learning a small, lightweight mapping network (e.g., a simple MLP) to bridge the gap? Was Procrustes chosen primarily for its simplicity and training-free nature, or did it also empirically outperform other simple alternatives?
> >
> > Thank you for your comment. We selected Orthogonal Procrustes primarily for its simplicity, training-free nature, and efficiency, and also because it performs competitively with trained linear mappings in low-data regimes. Prior work [A] shows that Procrustes performs on par with a learned linear layer and can even outperform it when only a few alignment samples are available. Since our pipeline relies on a minimal set of anchors, introducing a trained mapping network (e.g., a lightweight MLP) would not yield clear benefits and would diminish the appeal of a purely training-free formulation.
> >
> > [A] Maiorca et al., “Latent Space Translation via Semantic Alignment”, NeurIPS 2023.

---

> ### Comment · Reviewer_fJkf · 2025-11-26
>
> Thank you for addressing all my concerns and for providing evidence to clarify my questions and raiseed points. I will maintain my score

---

> > ### Author Response · Authors · 2025-11-27
> >
> > Thank you again for your thoughtful follow-up and for the time you dedicated to reviewing our work. We are glad the additional analyses clarified the points you raised. If you believe any further refinements during the revision window could strengthen the submission, we would be happy to incorporate them.
> >
> > We also note that a score of **6** is described as “marginally above the acceptance threshold, **but would not mind if the paper were rejected.**”
> >
> > If, after our clarifications and the additional evidence provided, you now feel the contribution merits stronger support, especially considering your evaluation of the paper’s presentation as **excellent** , contribution as **excellent**, and soundness as **good**, we would appreciate any adjustment you consider appropriate.
> > Otherwise, any remaining suggestions for improvement would be equally welcome.
> >
> > Thank you again for your careful review and for your constructive contributions to improving the paper.

---

### Official Review · Reviewer_kSFm · 2025-10-31

**Soundness:** 3
**Presentation:** 3
**Contribution:** 2
**Rating:** 6
**Confidence:** 3

**Summary:**

This paper investigates image synthesis through CLIP inversion. Rather than optimizing directly in the pixel space, they leverage implicit neural representations (INRs) to perform the inversion. To stabilize the inverse mapping, they incorporate techniques such as adversarial training, orthogonal Procrustes projection, and a blending loss.

**Strengths:**

- The paper provides a well-rounded review of related works across multiple relevant areas, helping readers grasp the broader context and motivation behind the proposed approach.
- It integrates several complementary techniques to tackle distinct challenges—for instance, using Adversarial Weight Perturbation (AWP) for robust INR initialization and incorporating natural image priors through blending.
- Unlike prior CLIP inversion methods that rely on direct pixel-space optimization and often produce noticeable artifacts, the proposed approach demonstrates improved visual quality in its qualitative results.
- The paper includes comprehensive ablation studies, analyzing the contribution of each introduced component both quantitatively and qualitatively.

**Weaknesses:**

- There are some minor issues in how certain points are presented. For instance, in the related work section, the statement “Modern image generators fall into three…” overlooks autoregressive models, which are also widely used and should be acknowledged.
- From a practical perspective, while the approach is conceptually interesting, it’s not particularly applicable in real-world scenarios. As shown in the results, the generated images still lag far behind other methods. However, the work remains valuable from an interpretability and analytical standpoint.
- The visualization and presentation could be improved. Specifically, Figures 5 and Tables 2 and 3 are rendered too small, which hinders readability.

**Questions:**

-

---

> ### Author Response · Authors · 2025-11-22
> **Answer to Reviewer kSFm**
>
> We appreciate the positive evaluation from `Rev. kSFm`, who comments on the comprehensive framing of prior work, the integration of complementary techniques (AWP, Procrustes, blending, FINER), and the clarity of the ablation studies. These strengths are also emphasized by `Rev. fJkf`, who highlights the originality and conceptual significance of exposing generative behavior within a frozen encoder through INRs. The interpretability focus noted by `Rev. kSFm` aligns with comments from `Rev. JCC3` and `Rev. YbCx`, who likewise view the method as a useful analytical tool for understanding CLIP’s embedding space. The practical limitations raised, primarily related to photorealism and runtime, are consistent across reviewers and represent a shared, well-defined set of concerns that we can address.
>
> ### W1 - Minor issue: missing autoregressive models in related work
>
> > W1) There are some minor issues in how certain points are presented. For instance, in the related work section, the statement “Modern image generators fall into three…” overlooks autoregressive models, which are also widely used and should be acknowledged.
>
> We thank the reviewer for pointing this out. Following your remark, we have now added in the revised manuscript in section 2, in the paragraph **image generative models**, further clarifying the suggested discussion.
> We have added the fourth class of generative models, which is autoregression (AR). In particular, we have discussed the impact that autoregressive models have had in the natural language processing domain, citing the seminal paper (Vaswani et al., 2017). Additionally, we have discussed how, even in the image domain, recent papers like Tian et al. (2024) demonstrate that AR models can compete with Diffusion Models in the image generation task by framing autoregressive learning on images as a “next-scale prediction” or “next-resolution prediction” approach.
>
>
> ### W2 - Applicability in real-world scenario, interpretability, and analytical value
>
> > W2) From a practical perspective, while the approach is conceptually interesting, it’s not particularly applicable in real-world scenarios. As shown in the results, the generated images still lag far behind other methods. However, the work remains valuable from an interpretability and analytical standpoint
>
> We thank the reviewer for acknowledging the important contribution of our work from an **interpretability and analytical standpoint**. This indeed represents a central contribution of our work.
>
> To further clarify this additional contribution, we have added a deeper discussion about this in the introduction of the paper, where we specify that CLIP$^{-1}$ is not intended to replace diffusion or autoregressive decoders, stating this clearly at the end of the section.
> Our goal is indeed different: CLIP$^{-1}$ is primarily an interpretability tool, which exposes how much image structure is already present in the frozen CLIP embedding space.
> As an example, CLIP$^{-1}$ supports uses that do not require a high-fidelity decoder. **In the new Fig. 5**, in the revised paper, we use inversion to visualize how CLIP reacts to negations (`“this is not a photo of…”`) and abstract OOD prompts, revealing which cues are amplified or collapsed and how its embedding geometry shifts. This makes the method useful for interpretability, stress-testing, and debugging multimodal systems before errors propagate into downstream VLMs and diffusion models.
>
>
> ### W3 - Minor presentation issues
>
> > W3) The visualization and presentation could be improved. Specifically, Figures 5 and Tables 2 and 3 are rendered too small, which hinders readability.
>
> We thank the reviewer for pointing this out. Following the suggestion, we used the additional space available in the rebuttal and camera-ready versions to improve clarity. Tables 2 and 3 are now properly reformatted as Table 3 (a,b,c), with larger fonts and a cleaner layout. The old Figure 5 on the ablation study has been moved to the appendix (Appendix A.7, **Figure 9**) and is now rendered at full page scale for better readability. Additional samples of ablations are presented in **Fig. 10** of the Appendix.

---

### Official Review · Reviewer_YbCx · 2025-10-31

**Soundness:** 3
**Presentation:** 3
**Contribution:** 2
**Rating:** 4
**Confidence:** 3

**Summary:**

This paper propose a new method of inversing CLIP encoder to achieve text-to-image geneartion using $\textnormal{CLIP}^{-1}$.
This is achieved thorugh a training-free approach, leveraging INR to achieve image embeddings similar to target text embeddings.

**Strengths:**

The technical contributions of this work are interesting. The authors propose techniques to: (1) enable a training-free approach that generates images without training any decoder, (2) mitigate the CLIP modality gap, and (3) achieve a robust, generalizable framework by leveraging AWP.

The comparison to existing work is solid, and the reported FID and other metrics are reasonable for this class of technique.

**Weaknesses:**

The primary concern with this work is its relevance and timeliness. The idea of inverting CLIP for image generation was novel and exciting a few years ago; however, the community has since shifted toward more scalable and expressive generative models.

Moreover, the outputs here are not yet high-quality enough, which is a significant shortcoming for practical use. Therefore, the work’s practical impact seems limited to me.

**Questions:**

How do the authors think about the scalability of this approach, given the per-image optimization loop?

Also, would this method extend to higher-resolution outputs?

---

> ### Author Response · Authors · 2025-11-22
> **Answer to Reviewer YbCx (part 1/3)**
>
> We thank the reviewer for highlighting the main technical contributions of our work, including the training-free inversion procedure, the modality-gap correction, and the robust INR initialization strategy. The reviewer also notes that **our comparisons against existing approaches are solid for this class of methods**. These observations are consistent with the assessments of reviewers `kSFm` and `fJkf`, who emphasize the complementary nature of the proposed components (AWP, Procrustes, blending loss) and the clarity of the ablation studies. The interpretability value noted by the reviewer also aligns with comments from `JCC3` and `kSFm`. We address the reviewer's concerns below.
>
> ### W1) Timeliness and relevance of CLIP inversion
>
> > W1) The primary concern with this work is its relevance and timeliness. The idea of inverting CLIP for image generation was novel and exciting a few years ago; however, the community has since shifted toward more scalable and expressive generative models.
>
> We appreciate the concern. It is true that the community's mainstream attention has shifted toward large diffusion and autoregressive generators. At the same time, the specific line of decoder-free CLIP inversion is not only active but still *underexplored*. All of the methods we benchmark against (CLIPAG (2023), CLIP-JEM (2024), EB-CLIP (2024), and DAS (2025)) are very recent, all being published within the last two years. This shows that researchers are still actively probing whether a frozen encoder such as CLIP contains latent generative structure worth exploiting. This perspective is also gaining traction more broadly: in a recent CVPR 2025 workshop talk, Kaiming He argued that recognition and generation are two sides of the same coin, and that generation can be viewed as inverting the recognition pathway. Under this view, exploring encoder inversion is not a detour but a possible direction for future generative architectures that use a single model for both recognition (text–image alignment) and synthesis (inverting the encoder) [A].
>
> Our view is that this line of research is in an *exploratory* phase, not a mature one, and that this is precisely why it should be pushed. A frozen encoder is an unusual generative backbone: it is cheap, massively pretrained, and extremely well understood, yet its inversion behaviour is nowhere near fully mapped. Our work addresses several aspects that earlier inversion attempts left open: robustness, coarse-to-fine control, explicit modality-gap correction, and natural-image anchoring. These changes materially raise the ceiling of what decoder-free CLIP inversion can do.
>
> We want to clarify our positioning with a concrete analogy. Some influential generative ideas began as exploratory work outside the dominant paradigm of their time. *Sohl-Dickstein et al., "Deep Unsupervised Learning using Nonequilibrium Thermodynamics" (ICML 2015)* was published in a period completely dominated by GANs. Yet, it laid the conceptual foundation for modern diffusion models, score-based methods, and flow matching. We definitely do not claim our work has a comparable impact, but we believe the example illustrates that *exploration* can be valuable even when the field's "exploitation" phase points elsewhere.
>
> In the same spirit, CLIP$^{-1}$ is not proposed as a competitor to SDXL, but as a step toward understanding and repurposing encoders as generators. The fact that the area is not yet saturated is, in our view, another reason to explore it.
>
> [A] Kaiming He - Towards End-to-End Generative Modeling - Workshop on Generative Models for Computer Vision @ CVPR 2025

---

> ### Author Response · Authors · 2025-11-22
> **Answer to Reviewer YbCx (part 2/3)**
>
> ### W2) Output quality and practical impact
>
> We agree that our images do not reach the fidelity of large diffusion or autoregressive models. Those systems rely on heavyweight decoders trained on billions of image–text pairs, whereas our method intentionally avoids any decoder training and keeps CLIP frozen. Within that constraint, the fair comparison is to other decoder-free inversion methods, not to SDXL. In that regime, CLIP$^{-1}$ provides significant gains: **on COCO, we cut DAS's FID in half (161.8 -> 72.5) and almost double its IS (5.7 -> 9.5), and qualitatively, we remove most of the structural artifacts that previously made CLIP inversion unstable**.
> Regarding practical use, we agree that the outputs are not yet suitable for high-fidelity synthesis. But *at the current state*, the method already has concrete utility in settings where photorealism is not the goal:
>
> - **Interpretability and safety analysis.** CLIP inversion shows how the encoder responds to prompts, including negations and out-of-distribution concepts, a task that does not require pixel-perfect synthesis. This supports debugging of multimodal pipelines and the identification of biases or misalignments before they propagate into larger systems such as diffusion models or large multimodal models. **In the revised paper, Fig. 5 introduces a new interpretability experiment that visualizes how CLIP behaves under negations** (`“this is not a photo of…”`) **and OOD prompts like** `"An abstract and unrecognizable object"`.
> - **Training-free manipulation.** Reconstruction, text-guided edits, and style transfer all operate with a single frozen encoder, making CLIP$^{-1}$ useful for lightweight editing, prototyping, and CLIP-centric workflows that cannot afford a full generative model.
> - **Front-end initialization for larger generators.** The inversion can act as a coarse semantic initialization that a diffusion model then refines. This improves alignment and reduces sampling ambiguity without any decoder training.
> - **Model-centric applications.** Many existing systems rely heavily on CLIP as their semantic backbone. Being able to invert that backbone directly provides a practical and straightforward way to inspect or manipulate CLIP's internal representations.
>
> We now clarify in the discussion that CLIP$^{-1}$ is not intended to replace state-of-the-art generators. Instead, it highlights the latent generative structure inside a frozen encoder and supports practical interpretability and low-resource tasks. Further improvements to the backbone or to natural-image regularization may expand its practical impact beyond the current state.
>
>
> ### Q1 - Scalability of the per-image optimization loop.
>
> > Q1) How do the authors think about the scalability of this approach, given the per-image optimization loop?
>
> Our method indeed uses a per-image optimization loop, but this is not unusual in inversion-based or iterative generative methods. Existing CLIP inversion approaches (e.g., DAS, CLIP-Inv) also run for hundreds of steps, and diffusion models themselves rely on a recurrent denoising process (typically 20–50 steps for modern samplers). In other words, recurrence at inference time is a shared property of this family of methods rather than a specific limitation of CLIP$^{-1}$.
>
> To make the scalability of our approach explicit, we added an ablation that varies the number of inversion steps. The results---**Table 3(c) in the revised paper**---show how performance scales as we shrink the optimization loop:
>
> | Method | steps | FID $\downarrow$ | IS $\uparrow$ | CLIPSIM $\uparrow$|
> | -------- | -------- | -------- | -------- | -------- |
> | DAS Ensemble | 400| 121.6 | 8.36 | 36.9 |
> | DAS ViT | 400 | 161.8 | 5.7 | 22.7 |
> | CLIP-inv ViT | 400 | 140.1 | 4.8 | **61.4** |
> |||
> | CLIP$^{-1}$ | 400 | **72.5** | $\underline{\text{9.5}}$ | $\underline{\text{38.6}}$ |
> | CLIP$^{-1}$ | **40 (new)** | $\underline{\text{107.1}}$ | **10.6** | 17.8 |
>
>
> Even at **40 steps** (10× fewer iterations), CLIP$^{-1}$ still outperforms all prior inversion-based methods in FID, and its IS actually improves. This indicates that the method degrades gracefully as computation is reduced, and can be run in a “fast mode” when needed.
>
> Long-term, the optimization loop is not a fundamental obstacle. INRs provide a stable mapping from embeddings to weights, enabling **amortization**. A small predictor network can be trained to estimate INR weights in a single forward pass, using our optimized INRs as targets. Borrowing from Computer Graphics, Initializations can also be meta-learned to get an ideal starting point with a single inference pass, which makes the inversion converge in even fewer steps [A]. This mirrors how NeRFs evolved from per-scene optimization to feed-forward radiance-field predictors.
>
> In summary: the per-image loop is comparable to existing inversion and diffusion pipelines, scales down well in practice, and can be amortized in future work if even faster synthesis is desired.

---

> ### Author Response · Authors · 2025-11-22
> **Answer to Reviewer YbCx (part 3/3)**
>
> ### Q2) Higher-resolution outputs
>
> > Q2) Also, would this method extend to higher-resolution outputs?
>
> Yes, the method extends well to higher-resolution inputs! INRs naturally support flexible spatial resolutions: given that INRs maps grid positions (resolution) to pixel space, we can implement the inversion to a higher resolution simply by manipulating the input grid of INRs.
>
> We have added **a new Figure 11 in Appendix A.8 Higher Resolutions** showing additional qualitative visualizations produced by the method to higher resolutions: these new images are synthesized at $448 \times 448$ size, which is $2\times$ than the original CLIP input. The comparison shows original, upscaled vs our INR-based native synthesis.
>
> [A] Tancik et al, "Learned Initializations for Optimizing Coordinate-Based Neural Representations", CVPR 2021.

---

### Official Review · Reviewer_JCC3 · 2025-11-04

**Soundness:** 3
**Presentation:** 3
**Contribution:** 3
**Rating:** 2
**Confidence:** 3

**Summary:**

In this paper, the author introduces CLIP−1, a novel approach for text-to-image synthesis by inverting CLIP embeddings without relying on a pretrained generative decoder or fine-tuning CLIP. The method leverages Implicit Neural Representations (INRs) optimized layer-by-layer in a frequency-aware manner, enabling coarse-to-fine image generation. Overall, this inversion method sheds light on how encoder models can potentially be used as image generations and also on the inner workings of CLIP. However, I believe that there's still a way to go for such methods to be used as pure image generators.

**Strengths:**

- The paper proposes a decoder-free and tuning-free approach for text-to-image synthesis, which is computationally efficient.

- The method supports zero-shot generalization to multiple tasks, including reconstruction, style transfer, and controlled edits, within a unified framework

- Extensive experiments on MS-COCO and Flickr30k demonstrate strong performance and also ablations showing improvements over other inversion methods.

**Weaknesses:**

- While the generated images are semantically aligned, fine-grained spatial details and textures occasionally exhibit distortions or artifacts. I understand that the authors are not competing with SoTA image generation methods, but a discussion on it would be fruitful.

- I am a bit apprehensive on the practical applications of the method beyond just understanding the CLIP space. How far is the performance (quantitatively) from pure image generators?

- While FID, IS, and CLIP-SIM are reported, additional perceptual metrics or user studies could strengthen the evaluation of visual quality.

**Questions:**

Overall, the paper makes a significant contribution by repurposing a frozen discriminative model for generative tasks, showcasing its potential for image synthesis. However, limitations in handling complex prompts which suggest areas for improvement and lack of comparisons to pure image generators draw me towards a reject. The work however opens promising directions for leveraging pretrained models in generative applications.

I urge the authors to answer questions about the practical utility of inversions with encoder models? How do they see such methods replacing pure image generators or how can such methods be used to improve the encoder capabilities itself. I believe these are unanswered questions.

---

> ### Author Response · Authors · 2025-11-22
> **Answer to Reviewer JCC3 (part 1/3)**
>
> We thank the reviewer for their thoughtful feedback. We appreciate their recognition of our contributions as **significant**: a decoder-free and tuning-free formulation, zero-shot generalization across tasks, and comprehensive ablations. These strengths are consistently mentioned by other reviewers. `Rev. kSFm` and `Rev. fJkf` highlight the same aspects, including the originality of operating in INR weight space and the effectiveness of simple, training-free components such as Procrustes and AWP. The interpretability focus noted by `Rev. JCC3` also aligns with observations from `Rev. YbCx` and `Rev. kSFm`, who emphasize the usefulness of CLIP inversion for understanding CLIP’s embedding structure. We address the reviewer's concerns below.
>
> ### W1) Artifacts and fine-grained details.
>
> > W1) While the generated images are semantically aligned, fine-grained spatial details and textures occasionally exhibit distortions or artifacts. I understand that the authors are not competing with SoTA image generation methods, but a discussion on it would be fruitful.
>
> We agree that our generations can still exhibit local distortions and imperfect textures, and we have clarified this limitation in the revised Discussion/Conclusion. As the reviewer correctly noted, our goal is not to compete with state-of-the-art diffusion or autoregressive generators, which rely on large, fully trained decoders, but to show that a *frozen* discriminative encoder already contains enough structure to support coherent text-to-image synthesis when combined with a lightweight INR.
>
> Within this “decoder-free inversion” regime, artifacts are already substantially reduced compared to prior CLIP inversion methods. As reported in Table 1, **CLIP$^{-1}$ achieves a FID of 72.5 vs 161.8 and an IS of 9.5 vs 5.7 for DAS on MS-COCO**, and qualitatively, our outputs in Fig. 3 exhibit far fewer structural glitches than CLIP-Inv and DAS. The remaining artifacts largely reflect two intrinsic limitations of CLIP inversion: (i) CLIP embeddings are many-to-one in pixel space, so fine-grained spatial details are under-constrained by the latent alone, and (ii) we deliberately avoid any learned pixel-space decoder or diffusion prior, so all regularization must come from the INR and CLIP-based losses.
>
> We now explicitly discuss these trade-offs in the paper: CLIP$^{-1}$ is best viewed as a decoder-free way to probe and repurpose CLIP's embedding space (and to enable low-resource text-to-image synthesis), rather than as a drop-in replacement for state-of-the-art generative models. We also outline in the revised conclusion that combining our inversion scheme with an explicit natural-image prior (e.g., a lightweight decoder or a diffusion-based projection step) is a promising direction for further reducing artifacts and sharpening high-frequency details.

---

> ### Author Response · Authors · 2025-11-22
> **Answer to Reviewer JCC3 (part 2/3)**
>
> ### W2, Q2) Practical applications.
>
> > W2) I am a bit apprehensive on the practical applications of the method beyond just understanding the CLIP space. How far is the performance (quantitatively) from pure image generators?
>
> > Q2) I urge the authors to answer questions about the practical utility of inversions with encoder models? How do they see such methods replacing pure image generators or how can such methods be used to improve the encoder capabilities itself. I believe these are unanswered questions.
>
> We now make this distinction explicit. CLIP$^{-1}$ is not positioned as a replacement for large text-to-image generators. Diffusion and AR models still achieve superior FID because they train a full latent-to-image decoder on billions of image–text pairs. Our goal is different: we show that a *frozen* CLIP already contains enough generative structure to enable coherent synthesis, editing, and reconstruction without training any decoder.
>
> Quantitatively, this naturally leaves a gap to full generators (e.g., SDXL's FID in the single digits vs our 72.5). Within the decoder-free setting, however, CLIP$^{-1}$ closes most of the gap: **we halve DAS’s FID and almost double its IS**. This makes the proposed method a practical tool when a full generative stack is unavailable or undesirable, such as in low-compute environments, interpretability work, and CLIP-centric analysis.
>
> Beyond text-to-image generation, CLIP$^{-1}$ supports several practical uses that do not rely on a high-fidelity decoder. In Fig. 5 of the revised paper, we use inversion to directly visualize how CLIP responds to negations (`"this is not a photo of..."`) and abstract out-of-distribution prompts like `"An abstract and unrecognizable object"`.
> These images show which cues CLIP exaggerates, which ones collapse, and how its embedding geometry behaves under distribution shift. This makes the method useful for interpretability, stress-testing, and debugging multimodal systems before errors propagate into downstream VLMs and Diffusion Models.
>
> In response to the reviewer’s question about improving the encoder, inversion provides concrete diagnostics: mapping embeddings back to pixels reveals the attributes that CLIP overemphasizes, the ones it suppresses, and the regions where the space becomes unstable. This information can guide data curation, regularization, and safety interventions. CLIP$^{-1}$ does not alter the encoder, but it clearly shows *where* it breaks and thus *how* it can be strengthened.
>
> Additionally, the same framework supports lightweight editing, style transfer, and reconstruction without requiring the training of a decoder. It can serve as coarse initialization for diffusion models, which can then refine the results. We added these points to the revised Discussion section.
>
>
> ### W3) Additional evaluation metrics.
>
> > W3) While FID, IS, and CLIP-SIM are reported, additional perceptual metrics or user studies could strengthen the evaluation of visual quality.
>
> We appreciate the suggestion. We use the same three metrics as all prior inversion baselines (FID, IS, CLIPSIM), ensuring a fair and directly comparable evaluation. As reviewer `YbCx` also notes, these metrics are standard for this class of methods, and our comparisons are solid.
>
> Beyond these metrics, we include extensive qualitative analysis, out-of-distribution experiments on multiple datasets, and detailed ablations that isolate the contribution of each component. Together, these provide a broad and balanced evaluation of visual quality and semantic alignment.
>
> Nevertheless, we are fully open to incorporating additional perceptual metrics if the reviewer has specific recommendations.

---

> > ### Author Response · Authors · 2025-11-22
> > **Answer to Reviewer JCC3 (part 3/3)**
> >
> > ### Q1) Handling complex prompts and lack of comparisons.
> >
> > > Q1) Overall, the paper makes a significant contribution by repurposing a frozen discriminative model for generative tasks, showcasing its potential for image synthesis. However, limitations in handling complex prompts which suggest areas for improvement and lack of comparisons to pure image generators draw me towards a reject. The work however opens promising directions for leveraging pretrained models in generative applications.
> >
> > We thank the reviewer for raising this point. We acknowledge the difficulty with very complex or compositional prompts and now state this limitation explicitly in the Conclusion. This behaviour reflects constraints of CLIP's embedding structure rather than the inversion mechanism itself.
> >
> > Regarding comparisons to full generative models: we address this point directly in our response to W2 / Q2, where we discuss the quantitative gap to diffusion and autoregressive systems and explain how decoder-free methods should be interpreted relative to them.
> >
> > Overall, we would like to remark that while the work does not aim to compete with state-of-the-art generators, it advances the frontier of decoder-free inversion and reveals new generative structure inside frozen encoders. Since the reviewer explicitly states that the paper opens promising directions for leveraging pretrained models in generative applications, we kindly ask them to reconsider their score so that this line of research can gain broader visibility and continue to develop.

---

### Author Response · Authors · 2025-12-02
**Author summary for the Area Chair**

Dear Area Chair and Reviewers,

We appreciate the reviewers’ careful evaluations. The reviews consistently highlight several strengths of the work.

`Rev. JCC3` notes that the paper *“makes a significant contribution”* and *“opens promising directions.”*  `Rev. fJkf` rates the **contribution as excellent**, underscoring the conceptual originality of exposing generative structure inside a frozen encoder.  `Rev. kSFm` highlights the clarity of the formulation and the effective integration of complementary components.  `Rev. YbCx` acknowledges the technical solidity of the method.

Several reviewers (`JCC3`, `YbCx`, `kSFm`) emphasize the interpretability value of CLIP⁻¹. In the revision, we clarified that the **primary goal of the work is to provide a tool for probing and stress-testing CLIP’s embedding space**: identifying biases, exposing failure modes, examining how CLIP handles negations and abstract or out-of-distribution concepts, and supporting debugging and alignment analysis **before** CLIP is used inside larger pipelines such as VLMs or diffusion models. This objective is orthogonal to competing with diffusion generators.

Reviewers also note that the ablation studies are clear and informative (`kSFm`, `fJkf`), and **presentation quality is rated excellent** by `fJkf`.

Below we list the added experiments and revisions addressing all reviewer concerns.


### Runtime, VRAM, and scalability (`fJkf`, `YbCx`)
Reviewers asked for explicit runtime, VRAM, and step-count comparisons against both inversion baselines and generative models.
We added:
- Full **RTX 4060** benchmark (Table 6):
  **CLIP⁻¹ (40 steps) = 7.8s**, faster than GLIDE (9.8s) and using less VRAM (3.2 GB).
- Detailed comparison versus DAS (Appendix A.2).
- New step-scaling study (Table 3c) showing CLIP⁻¹ remains superior to all prior inversion baselines even at 40 steps.

### High-resolution synthesis (`YbCx`)
The reviewer asked whether the method scales to higher resolutions.
We added:
- New **448×448** native INR synthesis examples (Appendix A.8), illustrating that the INR naturally supports higher-resolution grids without modifying the architecture.

### Robustness to initialization (`JCC3`, `fJkf`)
Reviewers asked how strongly the initialization constrains the result and how the method behaves with random or wrong seeds.
We added:
- **Wrong-seed experiments** (Appendix A.9): even semantically incorrect seeds converge to the correct semantic meaning.
- Enlarged and clarified AWP ablations (Appendix A.7–A.8), shown at higher resolution with supporting quantitative results (Tab. 3b).

### Interpretability and purpose of the method (`JCC3`, `YbCx`)
Reviewers asked for clearer positioning of the work’s purpose and practical value; we addressed this through a new interpretability experiment.
We added:
- **Negation and abstract-OOD** visualization experiments (Fig. 5), showing how CLIP behaves under distribution shift.

### Presentation and completeness (`kSFm`)
The reviewer asked for improvements in clarity, readability, and completeness of the manuscript.
We made the following changes:
- Expanded related work with autoregressive models.
- Enlarged Tables 2–3, and moved ablations to full-size appendix figures.

### Timeliness and relevance of the topic (`YbCx`)
The reviewer notes that the idea “was novel a few years ago,” though this seems at odds with the current literature: all comparable methods we evaluate against were published between 2023 and 2025, including CLIPAG (2023), CLIP-JEM (2024), EB-CLIP (2024), DAS (2025)

Thus, **decoder-free CLIP inversion remains an active and current research direction**, not a dated one.

### Final Note to the AC

The reviewers collectively recognize:
the **significance**, the **originality**, the **interpretability value**, the **technical solidity**, and the **clarity** of the work.

We provided substantial new experiments and revisions addressing every concern. `JCC3` and `fJkf` requested robustness tests (AWP, random seeds, wrong seeds); `fJkf` and `YbCx` asked for runtime, VRAM, and step-count evaluations; `YbCx` asked for higher-resolution results and scalability clarification; `JCC3` and `YbCx` asked for clearer positioning and practical purpose of the work, which we addressed with clarifications in the revised paper, along with new interpretability experiments, and finally `kSFm` requested minor presentation fixes and an update to related work. All are now addressed.

Thank you for your consideration.

---

### Meta-Review · Area_Chair_64se · 2026-01-07

**Summary:**

This paper proposes CLIP^{-1}, which uses an implicit neural representation (INR, pixel coordinates to RGB pixel) to invert the CLIP encoder for text-to-image synthesis. The reviewers generally appreciate the novelty and conceptual contribution of the work. The idea of using frequency-aware INR for modality gap correction.

**Reviewer Concerns:**

The major concerns are the method's practical utility and quality. Here is the summary

1. Image quality (JCC3, YbCx): Reviewers mentioned that the generated images still exhibit distortions and lack the fine-grained details of the pure generators.
2. Robustness to initialization (JCC3, fJkf): The reviewers asked how AMP-based seed constrained the output.
3. Resolution (YbCx): The reviewers asked whether the approach can produce a higher resolution image.
4. Inference time (fJkf, YbCx): The reviewers requested the computational cost and GPU RAM usage of each optimization loop.
5. Missing literature (kSFm): The reviewers requested to include autoregressive models in the related work section.

**Reviewer Scores:**

The initial score is derived as follows:

- JCC3: reject, not good enough
- YbCx: marginally below the acceptance threshold
- kSFm: marginally above the acceptance threshold
- fJkf: marginally above the acceptance threshold

The authors provided an extensive rebuttal to answer the reviewers' questions. Regarding 1 in the reviewers' concerns, the authors mentioned that the main purpose of this work is to interpretability and decoder-free probing. The authors also mention that the visual quality is significantly improved compared with prior work. Regarding 2, the authors added experiments in the appendix to show that even an incorrect seed can converge to the correct prompt meaning. Regarding 3, the authors provided new results with 448x448 pixels. Regarding 4, the authors mentioned that a new benchmark that validates faster speed and lower RAM usage.

Overall, AC confirms that the rebuttal can address reviewers' main concerns, and AC recommends acceptance of the paper. However, AC notes that "This decision can be bumped down" as the decision confidence. It is due to the practicality concerns raised by reviewers. Although the authors mentioned that the recent approach, such as DAS, has been published recently, it does not mean CLIP inversion is a long-standing problem. In that sense, AC agrees with reviewer YbCx's concerns.

---

### Decision · Program_Chairs · 2026-01-26

Accept (Poster)